# SYLBER: SYLLABIC EMBEDDING REPRESENTATION OF SPEECH FROM RAW AUDIO

**Cheol Jun Cho**[1], **Nicholas Lee**[1], **Akshat Gupta**[1], **Dhruv Agarwal**[1], **Ethan Chen**[1],
**Alan W Black**[2], **Gopala K. Anumanchipalli**[1]
[1]University of California, Berkeley, [2]Carnegie Mellon University

## ABSTRACT

Syllables are compositional units of spoken language that efficiently structure human speech perception and production. However, current neural speech representations lack such structure, resulting in dense token sequences that are costly to process. To bridge this gap, we propose a new model, Sylber, that produces speech representations with clean and robust syllabic structure. Specifically, we propose a self-supervised learning (SSL) framework that bootstraps syllabic embeddings by distilling from its own initial unsupervised syllabic segmentation. This results in a highly structured representation of speech features, offering three key benefits: 1) a fast, linear-time syllable segmentation algorithm, 2) efficient syllabic tokenization with an average of 4.27 tokens per second, and 3) novel phonological units suited for efficient spoken language modeling. Our proposed segmentation method is highly robust and generalizes to out-of-domain data and unseen languages without any tuning. By training token-to-speech generative models, fully intelligible speech can be reconstructed from Sylber tokens with a significantly lower bitrate than baseline SSL tokens. This suggests that our model effectively compresses speech into a compact sequence of tokens with minimal information loss. Lastly, we demonstrate that categorical perception—a linguistic phenomenon in speech perception—emerges naturally in Sylber, making the embedding space more categorical and sparse than previous speech features and thus supporting the high efficiency of our tokenization. Together, we present a novel SSL approach for representing speech as syllables, with significant potential for efficient speech tokenization and spoken language modeling.

## 1 INTRODUCTION

Self-supervised learning (SSL) approaches have been successful in learning speech representations that encode rich speech contents useful for diverse speech downstream tasks (Baevski et al., 2020; Hsu et al., 2021; Hu et al., 2024; Mohamed et al., 2022; Yang et al., 2021). In particular, speech tokens obtained by quantizing SSL features are receiving attention for understanding and generating spoken language (Lakhotia et al., 2021; Kharitonov et al., 2021; Hassid et al., 2024; Lee et al., 2022; Zhang et al., 2023). Substantial evidence suggests that SSL features are highly phonetic (Hsu et al., 2021; Cho et al., 2023; 2024a; Choi et al., 2024), which suggests that these quantized tokens are sub-phonemic units that densely tile the phonetic space (Sicherman & Adi, 2023). While capturing fine-grained speech contents, most existing speech tokenization approaches yield high frequency tokens (25-75 Hz), resulting in a long sequence of tokens to be processed. As prevailing attention based neural networks (Vaswani, 2017) have a quadratic cost with respect to sequence length, it becomes infeasible to process longer sequences with phoneme-level granularity.

A major bottleneck of the inefficiency in modeling spoken language is a lack of structure in current neural speech representations. Unlike text, there is no clear delimiter nor orthographic symbol in speech audio, which are crucial in efficient and scalable processing as evidenced in the text domain. However, human speech perception is structured as being segmented (Greenberg, 1998; Oganian & Chang, 2019; Gong et al., 2023) and categorical (Liberman et al., 1957; Pisoni, 1973; Pisoni &

---

Correspondence to: Cheol Jun Cho `<cheoljun@berkeley.edu>`, Gopala K. Anumanchipalli `<gopala@berkeley.edu>`

Lazarus, 1974). We argue that the machine representation of speech should resemble these cognitive structures to allow similar efficiency as text processing. A natural segmented structure of speech is a syllable, which organizes speech sounds in time (MacNeilage, 1998; Greenberg, 1998), and ideally, the embedding of a syllable should represent contents in a categorical way to be symbolized efficiently.

To this end, we propose a novel SSL framework that induces clean and robust syllabic structures in speech representations. Specifically, we build on top of a previous self-supervised syllable learning model, SDHuBERT (Cho et al., 2024b), and iteratively refine the syllabic segments that naturally arise from the model. Unlike the original model, which induces syllable structure as a byproduct of sentence-level SSL, we directly impose syllabic structures by regressing features against unsupervised syllable segments extracted from a teacher model which is initially set as the training model. We call the resulting model **Sylber** (**Syl**labic **e**mbedding **r**epresentation). [1]

The features from Sylber exhibit salient syllabic structure—showing a flat, consistent output within each segment and distinctive from other syllables (Figure 2, right). This enables a fast, linear time algorithm for segmenting these features. Moreover, this allows more accurate boundary detection and clustering that is more coherent with ground truth syllables than previous approaches. Syllabic tokens quantized from Sylber features show significantly lower frequency at an average of 4.27 token/second (Tok/s), and can be used to synthesize fully intelligible speech.[2] Furthermore, spoken language models based on syllabic tokens show comparable or better performance than the baselines with a similar resource setting, in learning lexicons and syntax.

To test whether Sylber is categorical, we probe the embeddings of a continuum of speech samples that interpolate rhyming word pairs, inspired by linguistics (Liberman et al., 1957). We introduce the Discriminability Index (DI) to quantify the degree of categorical perception of a speech representation model. Surprisingly, we observe a transient boundary drawn in the middle of the continuum, showing the best DI across SSL models. This suggests that the learned features are discretized in embedding space, contributing to the high efficiency of our syllabic tokens. To the best of our knowledge, this is the first demonstration of the validity and effectiveness of speech tokenization at the syllable level, with a tight connection to linguistic theories.

We summarize our contributions as follows:

- We propose **Sylber**, a novel SSL framework that imposes salient and robust syllabic structure in speech representation.
- Sylber outperforms previous approaches in syllable detection and discovery with a more efficient segmentation algorithm with $O(n)$ time complexity.
- The syllable segmentation by Sylber is generalizable to noisy conversational speech and even to unseen languages (Spanish and Mandarin) while being trained only on English audiobook
- We use this model to build a new dynamic speech tokenization scheme that has significantly lower sampling rate as **4.27** Tok/s on average, **6-7** times improvement over HuBERT tokens.
- We demonstrate that fully intelligible speech can be reconstructed from syllabic tokens, and that these units are suited for lexical and syntactic understanding.
- We demonstrate that categorical perception arises in Sylber, projecting audio to a more categorical embedding space than previous SSL models.

## 2 RELATED WORK

**Self-supervised learning in speech** Self-supervised learning (SSL) has been leveraged to learn rich representations from large, unlabeled speech corpora (Hsu et al., 2021; Baevski et al., 2020; Chen et al., 2022; Chung et al., 2021; Mohamed et al., 2022). Notably, HuBERT (Hsu et al., 2021) and WavLM (Chen et al., 2022) are pretrained using masked prediction on audio signals in order to extract representations on the audio for each frame. These SSL techniques typically extract representations at a fixed frame rate at 50 Hz, which is fairly finegrained and suggests that these repre-

---

[1]The code is available here: `https://github.com/Berkeley-Speech-Group/sylber`.
[2]Audio samples are at `https://berkeley-speech-group.github.io/sylber`.

sentations are highly correlated with sub-phonemic structures (Hsu et al., 2021; Cho et al., 2023; Abdullah et al., 2023; Sicherman & Adi, 2023; Baevski et al., 2021).

**Speech tokenization** Clustering and/or quantizing these SSL representations can provide speech tokens that are used for acoustic unit discovery (Hallap et al., 2022), speech recognition (Baevski et al., 2021; Chang et al., 2024), speech synthesis (Polyak et al., 2021; Hassid et al., 2024), language modeling (Lakhotia et al., 2021; Borsos et al., 2022; Hassid et al., 2024; Zhang et al., 2023), and translation (Lee et al., 2022; Li et al., 2023). However, these tokens severely suffer from high sampling rates, which makes the downstream models hard to scale, and struggle to learn long-range dependencies and higher-level linguistic structures due to the lack of explicit word boundaries and longer sequences. These caveats can be greatly improved by tokenizing speech at syllable-level granularity, yielding far shorter token lengths than the existing methods. For example, in English, the typical speaking rate is 4-6 syllables per second.

**Syllabic structure in speech SSL** Previous studies have demonstrated that syllabic structure can be induced by SSL (Peng et al., 2023; Cho et al., 2024b; Komatsu & Shinozaki, 2024). Peng et al. (2023) shows that syllabic structure in SSL features can be induced by jointly training with images and spoken captions. SDHuBERT (Cho et al., 2024b) demonstrates that such syllabic induction can be free of other modalities, with a sentence-level SSL. Komatsu & Shinozaki (2024) combined frame-wise distillation with speaker augmentation to derive syllabic segments. All these methods then utilize an agglomeration algorithm on top of the learned features to infer syllable boundaries, and extract syllable embeddings by averaging frames within detected segments. However, these previous studies induce syllabic structures in indirect ways, resulting in noisy syllable boundaries. Moreover, it is unclear whether the discovered syllables are valid speech representations or tokens. Here, our approach greatly improves the quality of the segments, and we first demonstrate the efficacy and validity of syllabic tokens through various experiments.

## 3 METHODS

### 3.1 SELF-SEGMENTATION DISTILLATION

Sylber is trained by a novel SSL framework, self-segmentation distillation, that imposes more explicit inductive bias of segment structure in feature representations by directly solving the speech segmentation problem (Figure 1). Specifically, we use SDHuBERT (Cho et al., 2024b) as a starting point, and leverage its unsupervised syllable segments as pseudo labels for segmentation. The target segment labels are continuous embeddings averaged across frames within each segment which are extracted from a teacher model. The teacher model is set as the initial student model and fixed during training, where we have two training stages—use the initial segments by SDHuBERT and its model weights; and update the teacher model and segments using the student model trained in the first stage.

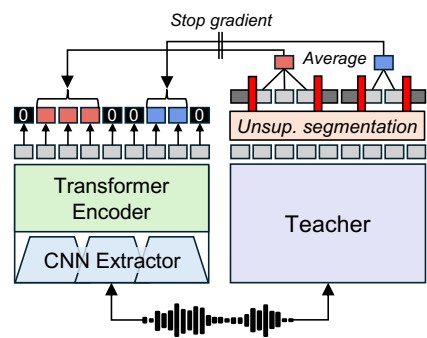

Figure 1: Self-segmentation distillation.

The loss objective is a frame-wise regression loss that minimizes the Mean Squared Error (MSE) between the output features at each frame and the target embeddings from the corresponding segment. Non-speech frames are regressed to zero, which are marked by norm thresholding by SDHuBERT (Cho et al., 2024b), and segments with low waveform amplitude (Appendix A.1.7). A formal definition of this distillation loss is described in Appendix A.1.1.

In addition to the distillation loss, we include a denoising objective similar to Chen et al. (2022) to improve robustness of the model, where 20% of the batch inputs for the student are mixed with environmental noise (Reddy et al., 2021) or other speech audio (Appendix A.1.3). This additional denoising is not a primary source of learning as a syllabic structure is readily visible without it, which is qualitatively shown in Appendix A.2.2. Furthermore, the model training is not sensitive to the choice of hyperparameters or model initialization (more discussed in Appendix A.2.6).

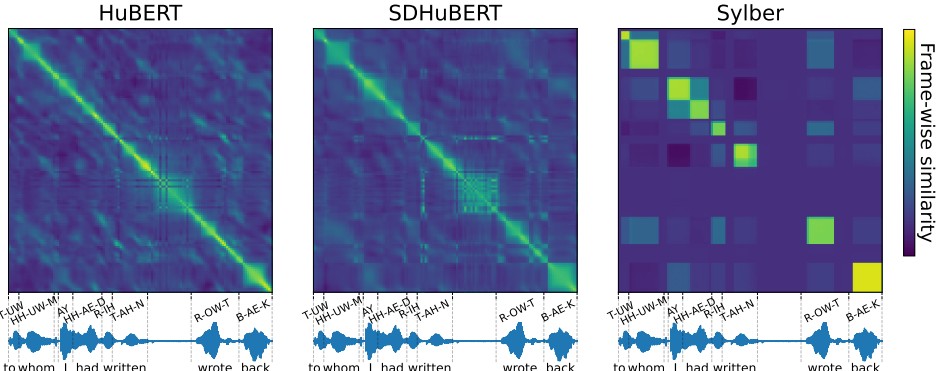

Figure 2: Frame-wise similarity matrix of raw features measured by dot product. For HuBERT and SDHuBERT, features from the ninth Transformer layer are extracted. Sylber shows extremely salient syllabic structure that is aligned with the ground truth syllable boundaries, with clear null activations in non-speech frames.

## 3.2 LINEAR TIME GREEDY SEGMENTATION ALGORITHM

The result of our self-segmentation distillation induces a framewise speech representation that exhibits a segmented structure as seen in the frame-wise similarity matrix (Figure 2, right). As we can see, our method produces a clean and robust segment structure that we can take advantage of to design a linear-time, greedy audio segmentation algorithm. The algorithm involves a monotonic agglomeration process where we sweep through each embedding and aggregate them into segments. While sweeping, the adjacent frames are merged together into a segment as long as their cosine similarity goes above a predefined merge threshold. The frames with L2 norm lower than a norm threshold are regarded as non-speech, and terminate agglomerating whenever encountered while sweeping. The greedy segmentation algorithm can sometimes make some errors by shifting some frames. Therefore, an additional pass is used to locally refine the boundaries by maximizing similarities between frames and assigned segments. See Appendix A.1.2 for details.

Each one of these steps can be implemented with $O(n)$ complexity, so the entire segmentation algorithm has linear complexity with respect to the audio sequence length. This allows fast online segmentation and is significantly more efficient than previous segmentation approaches (Peng et al. (2023); Cho et al. (2024b); Komatsu & Shinozaki (2024)) which all have $O(n^2)$ complexity as shown in Table 1. In Section 5.1, we show that this algorithm is on-par with previous $O(n^2)$ algorithms when applied to Sylber and other ablations involving previous models.

## 4 EXPERIMENTAL SETUP AND EVALUATION PROTOCOL

### 4.1 EXPERIMENTAL SETUP

**Architecture** Sylber has the same architecture as HuBERT with a CNN feature extractor followed by Transformer encoder. Based on the observation that the ninth layer of SDHuBERT best encodes syllables (Cho et al. (2024b)), we use a 9 layer transformer and initialize weights with SDHuBERT up-to that layer.[3] See Appendix A.1.5 for training details.

**Tokenization** To tokenize speech, we apply the aforementioned segmentation algorithm (Section 3.2) to get unsupervised speech segments. The features within segments are averaged to form continuous speech tokens at a syllable granularity (4-5 syllables per second). We apply a simple k-means clustering on the features with different vocab sizes (5K, 10K, and 20K). These cluster sizes are larger than what is used by other SSL-based clustering techniques (usually around 50-2K clusters), which is necessary since our features are closer to syllables than phonemes; similar to how vocabulary sizes for BPE based tokenizers are significantly larger than the number of characters.

---

[3]The checkpoint is retrieved from https://github.com/cheoljun95/sdhubert.

However, these syllabic tokens have a significantly lower temporal resolution compared to previous SSL-based tokens, which leads to improvements in efficiency (see Section 5.2).

**Token-to-speech** If our syllabic tokens are valid speech tokens, we should be able to reconstruct intelligible speech from them. We train a Conditional Flow-matching (CFM) (Lipman et al., 2022; Le et al., 2024) model to generate interpretable articulatory features that can be converted to speech audio using SPARC (Cho et al., 2024c). These articulatory features are speaker agnostic provided that pitch is normalized, while allowing full-reconstruction to speech. Since SSL-based speech tokens generally lack speaker information (Polyak et al., 2021; Wang et al., 2023), we aim to reconstruct these speaker-agnostic articulatory features from the syllabic tokens. See Appendix A.1.4 for the implementation and training details.

**Unit LM** Following Lakhotia et al. (2021), we train an autoregressive unit language model (uLM) using the syllabic tokens. The model has the same architecture as GSLM (Lakhotia et al., 2021), which is a decoder-only Transformer with 12 layers. Additional details are in Appendix A.1.6.

**Datasets** LibriSpeech (Panayotov et al., 2015) is used for training Sylber, and k-means clustering. For training the uLMs, we use either LibriSpeech or LibriLight (Kahn et al., 2020), and separately report performance. LibriTTS-R (Koizumi et al., 2023) is used for training the CFM models.

## 4.2 EVALUATION

**Syllable detection and discovery** We evaluate syllable boundaries with precision, recall, F1, and R-value following previous studies (Peng et al., 2023; Cho et al., 2024b; Komatsu & Shinozaki, 2024). Syllable discovery is evaluated by a separate clustering, where we use the same process as the previous works that use 4096 clusters. We measure syllable purity, cluster purity, and mutual information between the discovered syllables and the ground truth (Cho et al., 2024b; Komatsu & Shinozaki, 2024). See Appendix A.1.8 for details.

**Speech resynthesis** We measure reconstruction performance using the average Pearson Correlation of each component in articulatory features. To evaluate intelligibility, we use an off-the-shelf speech recognition model, Whisper (Radford et al., 2023)[4], and measure word error rate (WER) and character error rate (CER). Lastly, we apply an automated speech quality measurement, UTMOS (Saeki et al., 2022), to evaluate the quality of generated speech. These are evaluated on the test-clean split of LibriTTS-R. For some models, we collect subjective human evaluation on qualities, and report mean opinion scores (MOS) on the naturalness (nMOS) and prosodic similarity with the ground truth (psMOS).

**Coding efficiency** We evaluate the coding efficiency of the tokens with Token/second (Tok/s), bitrate, and coding-rate. The bitrate is calculated by $(\log_2(\text{vocab size})) \times \text{Tok/s}$. We define coding-rate as how much word information is preserved per bit: $\frac{(1-\text{WER}/100) \times \text{total \# of words}}{\text{total \# of bits}}$. Likewise, the test-clean split of LibriTTS-R is used to evaluate coding efficiency.

**Spoken Language Understanding** We use the zero-shot metrics of lexical learning, sWUGGY, and syntax learning, sBLIMP, following Lakhotia et al. (2021); Algayres et al. (2023). These metrics are measured by accuracy of discriminating real words/phrases and fake ones using the probabilities inferred from uLM, respectively.

### 4.2.1 BASELINES

For syllable detection and discovery, we compare our models against HuBERT, VGHuBERT, SDHuBERT, and Komatsu & Shinozaki (2024). For token-to-speech, we train the baseline CFM models using deduplicated HuBERT units with the size of 50, 100, and 200 by Lakhotia et al. (2021), and 500, and 2K by Nguyen et al. (2023); and SDHuBERT tokens with 5K, 10K, and 20K cluster sizes. For coding efficiency, we apply Byte Pair Encoding (BPE) using SentencePiece[5] to merge frequent units to form a larger vocabulary for HuBERT so that the size matches ours: 5K, 10K, and 20K, similar to Shen et al. (2024).[6] For evaluating language understanding, we use GSLM (Lakhotia

---

[4]We use "openai/whisper-large-v3" from Huggingface.

[5]https://github.com/google/sentencepiece

[6]The coding efficiency metrics are substantially worse using HuBERT without BPE due to their sampling granularity; thus, we compare against HuBERT with BPE to make a more fair comparison.

Table 1: Syllable detection and discovery results measured. Pr: precision, Re: recall, R: R-value, SP: syllabic purity, CP: cluster purity, and MI: mutual information. Complexity indicates time complexity of post-hoc segmentation algorithm. $n$: the number of frames and $k$: the number of syllables. All metrics are measured using LibriSpeech test data. Results of other *model–algorithm* combinations are also compared. Sylber uses a linear time algorithm while the other models use a quadratic time algorithm, which is only available with the clean structure learned by our approach.

| Model | Complexity | Syllable Detection | | | | Syllable Discovery | | |
|---|---|---|---|---|---|---|---|---|
| | | Pr↑ | Re↑ | F1↑ | R↑ | SP↑ | CP↑ | MI↑ |
| HuBERT | $O(kn^2)$ | 51.4 | 31.4 | 39.0 | 50.1 | 33.1 | 28.4 | 3.54 |
| VGHuBERT | $O(kn^2)$ | 65.3 | 64.3 | 64.8 | 70.0 | 53.4 | 43.6 | 4.66 |
| SDHuBERT | $O(n^2/k)$ | 64.3 | **71.0** | 67.5 | 70.7 | 54.1 | **46.2** | 4.76 |
| Komatsu & Shinozaki (2024) | $O(kn^2)$ | 73.3 | 67.6 | 70.3 | 74.6 | 59.4 | 44.5 | 5.08 |
| Sylber | $O(n)$ | **76.6** | 68.3 | **72.2** | **75.9** | **64.0** | 43.9 | **5.28** |
| Sylber–MinCut | $O(n^2/k)$ | 76.8 | 68.1 | 72.2 | 75.8 | 63.9 | 44.0 | 5.29 |
| HuBERT-Greedy | $O(n)$ | 54.5 | 35.2 | 42.8 | 52.7 | 29.5 | 25.9 | 3.36 |
| SDHuBERT-Greedy | $O(n)$ | 56.1 | 67.4 | 61.2 | 62.1 | 30.0 | 41.5 | 2.67 |

Table 2: Syllable detection performance in out-of-domain data. Sylber can generalize across novel domain and langauges without any tuning, showing high scores in all detection metrics.

| Corpus | Language | Style | Pr↑ | Re↑ | F1↑ | R↑ |
|---|---|---|---|---|---|---|
| LibriSpeech (in-domain) | English | Reading | 76.6 | 68.3 | 72.2 | 75.9 |
| Fisher | English | Conversation | 78.8 | 66.2 | 71.9 | 75.0 |
| MLS | Spanish | Reading | 73.5 | 69.9 | 71.7 | 75.9 |
| AISHELL-3 | Mandarin | Reading | 74.9 | 68.0 | 71.3 | 75.3 |

et al., 2021), tGSLM (Algayres et al., 2023), NAST (Messica & Adi, 2024), TWIST (Hassid et al., 2024) as baselines. These are selected as their tokenizers stem from HuBERT.

## 5 RESULTS

### 5.1 SYLLABLE DETECTION AND DISCOVERY

Table 1 shows a comparison of syllable detection and discovery performance. Sylber outperforms all previous approaches in every metric other than recall and cluster purity. As these two terms can be inflated by having more segments, it indicates that SDHuBERT is oversegmenting. In terms of discovery, we find the ground truth syllables are more purely mapped to ours than the baselines, greatly improving the previous SOTA by a huge margin (59.4 → 64.0). The results indicate that our model can detect and discover syllables better than the previous approaches. (Check Appendix A.2.7 for visualizations of the embedding space.) Moreover, the output features from our model are significantly cleaner than HuBERT or SDHuBERT as shown in Figure 2, which shows highly consistent similarities within syllable spans. This allows for a much faster $O(n)$ algorithm to be applicable, compared to the previous $O(kn^2)$ and $O(n^2/k)$ algorithms where $n$ is the number of frames and $k$ is the estimated number of syllables controlled by a hyperparameter. Compared to SDHuBERT, our syllable segmentation shows approximately $4\times$ gains in inference time (Appendix A.2.8).

When we apply the previous algorithm, MinCut (Peng et al., 2023; Cho et al., 2024b) to Sylber, the scores show very marginal differences (Sylber-Mincut in Table 1). MinCut is designed to search for optimal segments at the cost of computation time. Therefore, this indicates that Sylber features are clean and robust enough to find optimal segments in a greedy manner. In fact, when the greedy linear-time algorithm is applied to SDHuBERT, we find significant performance degradation (SDHuBERT-Greedy). In the case of HuBERT, both Greedy and MinCut show generally low performance due to the lack of syllabic structure.

Moreover, Sylber demonstrates surprising generalizability to unseen domain and languages without any tuning. As shown in Table 2, the syllable detection by Sylber shows high performance when applied to other corpora: conversational speech (Fisher; Cieri et al. (2004)), Spanish (MLS; Pratap

Table 3: Resynthesis results. HB: HuBERT, SDHB: SDHuBERT, and KM: KMean cluster size. Reconstruction metrics are average Pearson Correlation and WER and CER are reported in percentage (%). 95% confidence interval is reported for reconstruction and quality. Best scores are highlighted with bold font and best scores with quantization are underlined.

| Model | | Reconstruction | | | Intelligibility | | Quality | Frequency |
|---|---|---|---|---|---|---|---|---|
| Upstream | KM | Art↑ | Loudness↑ | Pitch↑ | WER↓ | CER↓ | UTMOS↑ | Tok/s↓ |
| HB | 50 | $0.926 \pm 0.065$ | $0.880 \pm 0.089$ | $0.586 \pm 0.581$ | 13.32 | 7.24 | $4.190 \pm 0.553$ | 23.59 |
| | 100 | $0.941 \pm 0.046$ | $0.878 \pm 0.098$ | $0.594 \pm 0.560$ | 7.78 | 3.89 | $4.177 \pm 0.548$ | 26.68 |
| | 200 | $0.944 \pm 0.044$ | $\underline{0.886 \pm 0.090}$ | $0.608 \pm 0.573$ | 6.34 | 3.10 | $4.197 \pm 0.543$ | 28.97 |
| | 500 | $0.941 \pm 0.043$ | $0.882 \pm 0.095$ | $0.623 \pm 0.532$ | 5.47 | 2.69 | $4.198 \pm 0.533$ | 29.46 |
| | 2K | $\underline{0.945 \pm 0.040}$ | $0.883 \pm 0.095$ | $0.660 \pm 0.458$ | $\underline{5.04}$ | $\underline{2.46}$ | $4.197 \pm 0.551$ | 33.62 |
| SDHB | 5K | $0.925 \pm 0.066$ | $0.872 \pm 0.089$ | $0.757 \pm 0.384$ | 9.88 | 5.40 | $4.140 \pm 0.660$ | |
| | 10K | $0.927 \pm 0.064$ | $0.879 \pm 0.083$ | $0.759 \pm 0.412$ | 9.25 | 4.99 | $4.173 \pm 0.600$ | 5.24 |
| | 20K | $0.930 \pm 0.061$ | $0.883 \pm 0.081$ | $\underline{0.784 \pm 0.373}$ | 8.63 | 4.62 | $4.180 \pm 0.609$ | |
| | ∞ | $\mathbf{0.959 \pm 0.035}$ | $0.948 \pm 0.042$ | $0.906 \pm 0.217$ | 4.94 | 2.56 | $4.190 \pm 0.552$ | |
| Sylber | 5K | $0.919 \pm 0.072$ | $0.877 \pm 0.091$ | $0.739 \pm 0.431$ | 8.70 | 4.48 | $4.189 \pm 0.607$ | |
| | 10K | $0.922 \pm 0.064$ | $0.876 \pm 0.088$ | $0.753 \pm 0.424$ | 8.07 | 4.28 | $4.155 \pm 0.624$ | $\underline{\mathbf{4.27}}$ |
| | 20K | $0.924 \pm 0.066$ | $0.882 \pm 0.084$ | $0.774 \pm 0.374$ | 7.95 | 4.06 | $\mathbf{4.210 \pm 0.547}$ | |
| | ∞ | $0.957 \pm 0.037$ | $\mathbf{0.950 \pm 0.045}$ | $\mathbf{0.918 \pm 0.216}$ | 4.88 | 2.42 | $4.199 \pm 0.539$ | |

Table 4: Coding efficiency comparison.

| Model | Token/second↓ | | | Bitrate↓ | | | Coding-rate↑ | | |
|---|---|---|---|---|---|---|---|---|---|
| | Vocab size | | | Vocab size | | | Vocab size | | |
| | 5K | 10K | 20K | 5K | 10K | 20K | 5K | 10K | 20K |
| HB50-BPE | 7.45 | 6.82 | 6.30 | 91.57 | 90.68 | 90.00 | 0.0283 | 0.0285 | 0.0287 |
| HB100-BPE | 14.78 | 14.40 | 14.10 | 181.56 | 191.37 | 201.46 | 0.0152 | 0.0144 | 0.0137 |
| HB200-BPE | 16.67 | 15.99 | 15.53 | 204.79 | 212.41 | 221.84 | 0.0136 | 0.0132 | 0.0126 |
| SDHB | | 5.24 | | 64.39 | 69.63 | 74.87 | 0.0253 | 0.0243 | 0.0234 |
| Sylber | | **4.27** | | **52.43** | **56.70** | **60.97** | **0.0315** | **0.0302** | **0.0289** |

et al. (2020)), and Mandarin (AISHELL-3; Shi et al. (2021)), which are on par with the in-domain case (LibriSpeech). This multilingual generalizability indicates that Sylber represents phonological information which may have a shared physical basis across languages (Ohala, 1984; 1990; Cho et al., 2024a). This finding suggests a potential scalability of Sylber. More details and discussion are in Appendix A.2.5.

## 5.2 RESYNTHESIS PERFORMANCE AND CODING EFFICIENCY

The results of token-to-speech resynthesis are shown in Table 3 and can be heard here. We find a general trend in both SDHuBERT and our syllabic tokens that articulatory reconstruction and intelligibility increase with finer clustering granularity. For intelligibility, our model outperforms SD-HuBERT at every vocab size while requiring less tokens per second. Interestingly, the articulatory reconstruction is generally higher in SDHuBERT, but also less intelligible. This indicates that our model marginalizes out some amount of articulatory variance which is orthogonal to orthographic contents. This marginalization is also happening in intonation when the embeddings are quantized, as shown in the huge reduction in pitch correlation compared to non-quantized model, resulting in a flattened speech generation. This pattern is shared in both SDHuBERT and Sylber.

The articulation is slightly better reconstructed by HuBERT units with 100 or more clusters than the units from SDHuBERT or our model, as shown in Table 3. This can be attributed to HuBERT's sub-phonemic, articulatory representations (Cho et al., 2023), which have a fine temporal granularity of 26.68 Tok/s or higher. The intelligibility is also slightly better with HuBERT units of 100 or more clusters, with WERs of 5.04–7.78, compared to the best-performing syllabic units with a WER of 7.95. Though the difference is marginal, HuBERT units require at least 6-8 times more tokens per second. Also, we find HuBERT units perform worse in representing pitch, corroborating findings by Polyak et al. (2021); Kharitonov et al. (2021); Nguyen et al. (2023).

Table 6: Speech uLM performance comparison. Sections are divided by training data size (top: LibriSpeech (LS) and bottom: LibriLight (LL) or more).

| Model | # Param. | Vocab size | Tok/s↓ | Bitrate↓ | Corpus | Data size (hr) | sWUGGY↑ | sBLIMP↑ |
|---|---|---|---|---|---|---|---|---|
| GSLM | 150M | 100 | 26.68 | 177.26 | LS | 1K | 68.70 | 57.06 |
| SDHuBERT-uLM | 125M | 5K | 5.24 | 64.39 | LS | 1K | 65.80 | 54.87 |
| | | 10K | | 69.63 | LS | 1K | 67.42 | 54.48 |
| | | 20K | | 74.87 | LS | 1K | 67.85 | 54.87 |
| Sylber-uLM | 125M | 5K | 4.27 | **52.47** | LS | 1K | 67.32 | 57.34 |
| | | 10K | | 56.74 | LS | 1K | 68.41 | **58.04** |
| | | 20K | | 61.01 | LS | 1K | **70.27** | 57.67 |
| tGSLM | 150M | – | 5 | – | LL | 6K | 68.53 | 55.31 |
| NAST | 150M | 200 | 28.97 | 221.44 | LL | 6K | 76.42 | 55.62 |
| TWIST-ColdInit | 125M | 500 | 16.78 | 150.45 | LL++ | 150K | 77.74 | 54.27 |
| TWIST | 13B | 500 | 16.78 | 150.45 | LL++ | 150K | **84.10** | 59.20 |
| Sylber-uLM | 125M | 20K | **4.27** | **61.01** | LL | 66K | 76.31 | 60.54 |
| Sylber-w/SIL-uLM | 125M | 20K | 4.76 | 68.01 | LL | 66K | 78.03 | **60.78** |

To further evaluate coding efficiency, we compare Sylber against baselines with comparable settings of HuBERT units in Table 4. Our model outperforms each baseline in every metric, showing about a 20% gain over the SDHuBERT tokens. In addition, Table 4 demonstrates the innate inefficiency in previous approaches using HuBERT units. There is a minimal gain in sequence compression while increasing the vocabulary size, where BPE is not able to reduce Tok/s by even half of the original when applied to 100 and 200 clusters. The only comparable baseline is BPE on 50 HuBERT clusters, which can reduce Tok/s from 23.59 to between 6.30-7.45. However, there is a huge information loss as shown in the high WER of 13.32, which results in a lower coding-rate (0.0283, 0.0285, 0.0287) compared to ours (0.0315, 0.0302, 0.0289) for vocab size of (5K, 10K, 20K) respectively.

HuBERT units and Sylber units show comparable quality in terms of naturalness in both machine and human evaluation (UTMOS in Table 3; nMOS in Table 5). In terms of prosody, Sylber 20K units show higher subjective similarity than HuBERT 200 or 2K units as shown in the psMOS results (Table 5).

Without quantization, the best performance is achieved by our model, with a WER of 4.88, Tok/s of 4.27, and higher correlations in loudness and pitch (Table 3). Furthermore, both of the subjective qualities, nMOS and psMOS, significantly increase (Table 5). This indicates the significant potential of syllabic tokens as an efficient speech coding that can be harnessed by a better quantization method. We leave this investigation for future work.

Table 5: Subjective evaluation on resynthesis quality.

| Model | KM | nMOS↑ | psMOS↑ |
|---|---|---|---|
| GT | | 4.37 | 4.71 |
| HB | 200 | 3.24 | 2.65 |
| HB | 2K | 3.33 | 2.90 |
| Sylber | 20K | 3.32 | 3.04 |
| Sylber | ∞ | **3.80** | **3.62** |

## 5.3 Spoken Language Understanding

Table 6 compares the sWUGGY and sBLIMP scores of speech uLMs. In the limited resource setting that uses 1K hours of training data. Sylber-uLMs generally outperform the baselines, GSLM and SDHuBERT-uLMs at sBLIMP, and the model with 20K vocab size outperforms GSLM in sWUGGY, while none of the SDHuBERT-uLMs outperform GSLM or Sylber-uLMs (top section of Table 6). This indicates that our syllabic tokens have better utility in terms of language modeling compared to the syllabic tokens from SDHuBERT. We also observe a general trend shared in SDHuBERT and ours that a larger vocab size yields a higher sWUGGY score, indicating that a finer clustering better covers the lexical space. Notably, Sylber-uLM trained on 1K hours outperforms tGSLM which has a similar token granularity as 5 Hz but with a fixed pooling window, even though their model is trained on a larger dataset of 150K hours. This suggests that a variable pooling window by syllabic segmentation is more suitable than using a fixed pooling window.

When we scale up the train data to 66K hours, Sylber-uLMs are able to achieve comparable or better sWUGGY scores than the models with similar sizes: tGSLM, NAST, and TWIST-ColdInit (bottom section of Table 6). We also include a silence interleaved version (Sylber-w/SIL-uLM), where we insert a silence token when the gap between two adjacent tokens is longer than 140 ms.

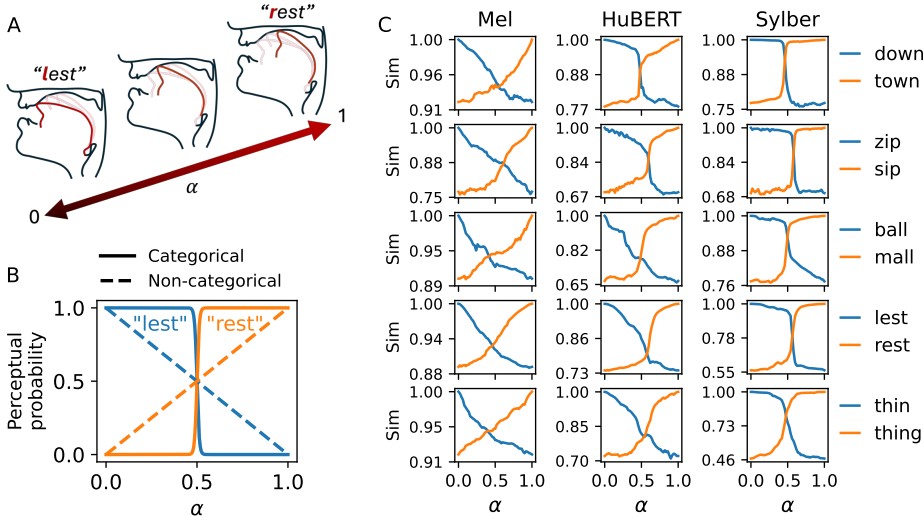

Figure 3: A. Overview of articulatory interpolation of rhyming words when interpolating $\alpha \in [0, 1]$. B. Hypothetical curves of categorical (solid lines) and non-categorical (dashed lines) embeddings. C. Similarity curves examples from Melspectrogram (Mel), HuBERT, and Sylber. Sylber consistently shows highly categorical perception, drawing a sharp boundary in continuum between words.

This improves the performance, especially sWUGGY, while increasing Tok/s and bitrate to 4.76 and 68.01, which are still far below the previous models. Although sWUGGY falls short of the TWIST model with 13B parameters, Sylber-uLMs show slightly better performance in sBLIMP, which is astonishing given the huge gap in model size and training data. Most importantly, all these results are obtained using significantly smaller sequence length and bitrate, several times lower than previous approaches. This suggests that Sylber units are highly efficient and valid tokens for spoken language modeling.

## 6 EMERGENT CATEGORICAL PERCEPTION IN SYLBER

The results shown in Section 5.2 suggest that HuBERT features might densely tile the embedding space, requiring many clusters to fully cover the space with fidelity. This lack of discreteness leads to a redundant token vocabulary (Sicherman & Adi, 2023), resulting in an exponentially growing token vocabulary size when tokens are grouped to reduce sequence length. However, human speech perception exhibits categorical effects, where listeners tend to impose discrete boundaries on a continuum of speech sounds rather than tracking gradual acoustic changes (Liberman et al., 1957; Pisoni & Lazarus, 1974; Harnad, 2003). This phenomenon—*categorical perception*—suggests that phonemic categories are perceived as discrete units despite underlying acoustic variability. If similar categorical effects emerge in the embedding space of SSL models, they could enable more efficient tokenization with a compressed set of discrete units. Based on this linguistic theory, we evaluate the degree of categorical perception in Sylber and other SSL models to assess the discreteness of their embedding spaces.

We simulate interpolation between two rhyming words to probe the embeddings of speech SSL models to check whether they are categorical. Specifically, monosyllabic words are used where a single consonant is different at the front (onset) or back (coda) of the syllable. We make the contrast to be varying one of the phonological properties: nasality, voicedness, or place (e.g., "**b**all" vs "**m**all", "**d**own" vs "**t**own", or "**l**est" vs "**r**est", respectively). We do not include vowel contrasts since categorical perception of vowels is not as consistent as consonants (Pisoni, 1973; Pisoni & Lazarus, 1974). We consider 13 types of such difference and simulate 4 pairs for each type, resulting 52 word pairs in total. The details of the difference types and the full list of word pairs can be found in Appendix A.1.9. To simulate a continuum between words, we utilize SPARC (Cho et al., 2024c) which allows direct editing in the physical articulatory space (Figure 3-A). We first generate audio

using the an off-the-shelf TTS API.[7] We extract articulatory features from the speech, which are then temporally aligned by dynamic time warping to either end if necessary. We sample 51 equidistant samples in the linear interpolation between two words, where each end is manually adjusted to make the perceptual boundary drawn approximately in the middle ($\alpha = 0.5$), which can be heard here. The pitch and loudness are also controlled to be at the same level. More details about SPARC can be found in Appendix A.1.4.

Given a speech representation model, we extract features for each interpolating point between words in each pair. We calculate the similarity between interpolating features with features from either end, forming a likelihood curve along the interpolation. Hypothetically, if the representation is categorical, the likelihood curves should show a sharp transition at the boundary (Figure 3-B). If the embeddings are not categorical and tracing the interpolation, the curves would show "X" pattern as the dashed line in Figure 3-B. We define the Discriminability index (DI) to quantify the level of categorical perception. The DI measures an empirical risk of wrong discrimination, where the probability is calculated based on similarities. See Appendix A.1.10 for the detailed definition. If the embeddings are categorical, DI will be close to 0. If they are non-categorical with X-shaped curves, DI will be 0.25. The maximum value of DI is 0.5, which would be random chance discrimination.

Figure 3-C illustrates that a clear boundary is drawn when interpolating between two rhyming words, whereas such a boundary is less prominent in HuBERT. This indicates that Sylber needs only 2 categories to represent the interpolating continuum, while HuBERT requires multiple categories or units, which induces high levels of inefficiency and redundancy. The similarity curves using the melspectrogram resemble an X-shape, indicating a non-categorical embedding space.

We compare Sylber with traditional acoustic features (Melspectrogram and MFCC), representative frame-wise SSL models (HuBERT (HB), Wav2Vec2 (W2V2), WavLM, and large (-L) versions (if applicable), and SDHuBERT. As shown in Table 7, our model's embeddings demonstrate the best discriminability, with the lowest DIs across both onset and coda contrasts (overall DI: 0.112). The results are unexpected as we only impose the model to learn temporal structure, and our loss objective does not involve any categorical learning at all. However, the embedding space of Sylber is naturally structured to be categorical, indicating the self-segmentation distillation might be a natural learning algorithm that resembles human language learning. Taken together, these qualitative and quantitative results suggest that the embedding space of Sylber is readily quantized, contributing to the performance improvements observed in previous sections.

Table 7: DI comparison.

| Model | DI↓ | | |
|---|---|---|---|
| | Onset | Coda | All |
| Mel | 0.198 | 0.193 | 0.196 |
| MFCC | 0.191 | 0.182 | 0.188 |
| W2V2 | 0.172 | 0.178 | 0.174 |
| HB | 0.136 | 0.152 | 0.141 |
| W2V2-L | 0.138 | 0.156 | 0.143 |
| HB-L | 0.166 | 0.180 | 0.170 |
| WavLM-L | 0.136 | 0.148 | 0.140 |
| SDHB | 0.133 | 0.126 | 0.131 |
| Sylber | **0.116** | **0.103** | **0.112** |

## 7 CONCLUSION

We propose a novel self-supervised learning framework of speech, Sylber, that learns to transform speech waveform into a syllabic embedding that is well aligned with linguistic theories. Sylber offers promising potential for interpretable and efficient speech tokenization, and scalable and efficient spoken language modeling.

**Limitations** As we present our model more as a *coding* framework of speech, we largely put our focus on demonstrating efficiency and reconstruction quality. Therefore, our model is not yet suitable for universal speech representation, which the most speech SSL approaches aim for (Yang et al., 2021). We find that Sylber degrades in some SUPERB downstream tasks, which we believe is due to the parsimonious structure we are imposing. See Appendix A.2.4 and Table 12 for details and discussion. Also, the SUPERB protocol is optimally designed for frame-wise SSL; therefore, more investigation is needed on downstream architectures that better leverage the syllabic structure.

---

[7]We use the TTS service in Vertex AI (https://cloud.google.com/vertex-ai) with a default female voice.

## ACKNOWLEDGEMENTS

This research is supported by the following grants to PI Anumanchipalli — NSF award 2106928, BAIR Commons-Meta AI Research, the Rose Hills Innovator Program, UC Noyce Initiative at UC Berkeley, and Google Research Scholar, JP Morgan AI research Award. Special thanks to Shang-Wen (Daniel) Li and Abdelrahman Mohamed for valuable discussions and advice.

## ETHICS STATEMENT

We believe that Sylber is a substantial step forward for speech models and spoken language understanding. Our technique enables efficient and effective speech tokenization which can potentially be used for malicious purposes. It is important for users, researchers, and developers to use this model and this framework ethically and responsibly.

## REPRODUCIBILITY STATEMENT

In the spirit of open research, we will be releasing all of the code associated with Sylber. We will release the pretrained model weights as well as the code necessary to retrain the model. In addition, we will be releasing all of the interpolation samples so that other researchers can also use our Discriminability Index as an evaluation metric for future research.

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

## A APPENDIX

### A.1 IMPLEMENTATION DETAILS

#### A.1.1 SELF-SEGMENTATION DISTILLATION

Given a speech audio signal $x$, we extract features, $M_S(x) = z^S$ and $M_T(x) = z^T$, where $M_S$ and $M_T$ are the student and teacher models, respectively. The unsupervised segmentation algorithm, **Useg**, outputs segment boundaries from $z$ as $\text{Useg}(z) = \{s\}^N$, where $N$ is the number of discovered segments, and $s \in \mathbb{N}^2$ denotes start and end frames of the segment, indexed as $s_{j,0}$ and $s_{j,1}$ for the $j$-th segment. We define an assignment function, $A(i) = j$, that gives the index of the segment, $j$, given a frame number, $i$, such that $s_{j,0} \leq i \leq s_{j,1}$. When there is no assignable segment, $A(i) = -1$, meaning $i$ is a non-speech frame. The segment-averaged feature, $v_j$, is defined by averaging across frames in the $j$-th segment: $v_j = \frac{1}{p-q} \sum_{k \in [p,q]} z_k$, where $(p,q) = (s_{j,0}, s_{j,1})$. $v_{A(i)}^T$ is defined as the teacher's segment-averaged feature of the segment that $i$-th frame belongs to, which is the target of the regression, which is zero for any non-speech frame: $v_{-1}^T = 0$. Finally, the loss function of the proposed self-segmentation distillation is defined as $\mathcal{L}_{\text{SegDistill}} := \sum_i ||v_{A(i)}^T - z_i^S||_2^2$.

#### A.1.2 GREEDY SEGMENTATION ALGORITHM

**Algorithm 1** Greedy Segmentation Algorithm

```
1:  procedure GREEDY-SEGMENTATION(states, N_thr, M_thr)
2:      Compute L2 norms and mark speech frames: speech_i = (||s_i||_2 ≥ N_thr)
3:      Initialize empty list of segments S
4:      for i = 1 to n do
5:          if speech_i and (no current segment or sim(s_i, avg(S_k)) < M_thr) then
6:              Start new segment S_{k+1} with s_i
7:          else if speech_i then
8:              Add s_i to current segment S_k and update avg(S_k)
9:          else if current segment S_k exists then
10:             Finalize current segment S_k
11:         end if
12:     end for
13:     for each boundary j between segments S_k and S_{k+1} do
14:         if speech_j then
15:             if sim(avg(S_k), avg(S_{k+1})) ≥ M_thr then
16:                 Merge S_k and S_{k+1}
17:                 Continue
18:             end if
19:             Define local search range from a = midpoint(S_k) to b = midpoint(S_{k+1})
20:             Get similarity to left or right segment L = [sim(s_i, avg(S_k))]_a^b, R = [sim(s_i, avg(S_{k+1}))]_a^b
21:             Find optimal boundary j* = arg max_j (sum(L_{1:j-1}) + sum(R_{j:b-a}))
22:             Update boundary to j*
23:         end if
24:     end for
25:     return S
26: end procedure
```

The algorithm involves three linear passes through the audio embeddings (Algorithm 1). The first step thresholds all embeddings based on their L2 norm, distinguishing between speech and non-speech segments. Next, a monotonic agglomeration process iterates through the embeddings, group-

ing them into segments. At each time step, a frame is merged into the current segment if its cosine similarity with the segment's average embedding exceeds a predefined threshold. This process runs in a single pass without constructing the entire similarity matrix by greedily starting a new segment whenever a frame falls below the threshold.

However, this greedy approach can introduce segmentation errors by slightly misaligning frame assignments. To correct this, a third pass refines segment boundaries. If adjacent segments have a cosine similarity above the merge threshold, they are merged. Otherwise, a local search range is defined from the midpoint of the previous segment to the midpoint of the following segment. Within this range, we compute the cosine similarity between each frame and the averages of the two segments. Instead of updating the averages dynamically, we use the original segment embeddings to keep the computational complexity linear. Finally, we determine the optimal boundary by maximizing the sum of cosine similarities between each frame and its assigned segment across all frames within the search range.

The time complexity is determined by the asymptotic number of dot product operations (or similarity computations), as these are the most computationally expensive steps in the segmentation algorithm.

### A.1.3 Noise Augmentation

For the denoising objective, we mix the input with a randomly sampled environmental sound or other speech audio. For mixing with environmental sound, we randomly select a clip from Reddy et al. (2021) and sample a 5 second clip from it. We first z-score the waveform and multiply by a factor sampled from $[0.05, 0.7]$, and mix with the original speech audio. Note that the original speech is also z-scored. For mixing with other speech, we randomly select another clip in the batch and shift from left or right with a percentage sampled from $[0.4, 0.7]$, to make sure the original speech holds the dominant information context in the mixture. The magnitude is also modulated by multiplying by a factor sampled from $[0.0, 0.2]$. We apply this augmentation to 20% of the samples in the batch, and only to the inputs fed to the student model. Within the 20%, we have the source of noise be 75% environmental noise and 25% other speech.

### A.1.4 Token-to-speech

**SPARC** The target of our token-to-speech model is processed by SPARC (Cho et al., 2024c), which is composed of articulatory encoding and decoding. The encoding pipeline outputs 14 articulatory features at 50 Hz, which are composed of the XY coordinates of 6 articulators (lower incisor; upper and lower lips; tongue tip, blade and dorsum;), loudness, and pitch. These are interpretable and grounded representations of speech that are fully informative of speech contents (Cho et al., 2024c). The decoder, or articulatory vocoder, is a Hifi-GAN (Kong et al., 2020) conditioned on a speaker embedding inferred from a separate speaker encoder. Cho et al. (2024c) shows that SPARC successfully decomposes speech contents and speaker identity, by normalizing pitch to remove speaker specific pitch level. We replicate the implementation from Cho et al. (2024c), except that we change the layer of WavLM in the speaker encoder from the CNN outputs to the sixth Transformer layer, based on the observation that this layer contributes the most to the downstream speaker identification task (Chen et al., 2022). Following Kharitonov et al. (2021), the pitch is normalized by dividing it by the mean pitch and then taking its logarithm for the target of the token-to-speech model. The original pitch range is restored from the predicted normalized pitch by reversing the normalization process. For training SPARC, we use LibriTTS-R using a single A5000-24GB GPU.

**Conditional flow-matching (CFM)** The input model in the CFM is composed of two feed forward networks (FFNs) and a linear layer, where each FFN has two linear layers with 512 hidden units and residual connection, with a ReLU activation and dropout rate of 0.05. Layernorm is applied to the output of each FFN. The final linear layer projects the 512 dimensional feature to 256. The Transformer in the CFM has 8 layers and each layer has 8 heads with 64 dimensions, and 512 for the encoding dimension. We use Rotary positional embeddings (Su et al., 2024). The final output is projected to the 14 dimensional flow in articulatory feature space. For training, the learning rate is fixed as 1e-4, with a batch size of 64 and 200k updates using a single A6000-48GB GPU.

**Inference** The SSL token embeddings are restored by the k-means codebooks, and expanded to the durations of the original segments. The non-speech frames are filled with zeros. For the case without quantization, the segment-averaged features are used. We use the extracted speaker embedding and

mean pitch from the original speaker to synthesize speech from articulatory features predicted from tokens. We use the original segment durations for each token without predicting them since the duration information can be easily tokenized. (For example, duration can be tagged for each token. See Appendix A.2.3.) We remove randomness in the CFM to yield consistent generations for the evaluation purposes.

### A.1.5 SYLBER TRAINING DETAILS

We train Sylber in two stages. The first stage is training with segment boundaries inferred from SDHuBERT which are extracted once at the beginning and fixed while in this stage. The second stage utilizes online segmentation using the teacher model's outputs, by the algorithm in Section 3.2. Note that this training is only possible since our model exhibits features clean enough for our greedy segmentation to work. In the second stage, the L2 norm threshold is updated online by aggregating the statistics of speech and non-speech segments, and the merge threshold is randomly sampled from $[0.8, 0.9]$. After the training, the norm threshold is fixed at 3.09 and the merge threshold is fixed at 0.8. See Appendix A.1.7 for details about the thresholding. The first-stage teacher is set and fixed as the initial student model, which is SDHuBERT with nine encoder layers. The second-stage teacher is then updated and fixed using the student model trained in the first stage. The model is trained for 115K steps in the first stage and further trained for 50K steps in the second stage. We use a batch size of 64 and each data point is randomly cropped to be 5 seconds, following Cho et al. (2024b). We show in Appendix A.2.9 that this does not significantly impact resynthesis for longer segments. The learning rate is set as 1e-4 with initial 500 warmup updates for the first stage and 5e-5 for the second stage. The second stage training improves performance in syllable detection and discovery (Appendix A.2.1). We used a single A6000-48GB GPU for training.

### A.1.6 ULM TRAINING DETAILS

For training uLMs, we largely follow Hassid et al. (2024). When training on LibriLight, we use 96% of the data for training and 2% each is held out for validation and test. For each speech clip, we sample 100 tokens for training, which is roughly corresponding to 20 seconds. We use a single A6000-48GB GPU for training uLMs on LibriSpeech and two of them for training on LibriLight.

### A.1.7 THRESHOLDS SETTING

**Thresholds in SDHuBERT segmentation** For segmenting SDHuBERT features, we apply the minimum cut algorithm (MinCut) introduced by Peng et al. (2023) and modified by Cho et al. (2024b). Following Cho et al. (2024b), the initial mask is obtained by thresholding norms of features from the eleventh layer of the Transformer, where we normalize norms to be in $[0, 1]$ and use 0.1 as threshold. The minimum cut refines each masked chunk to make it syllabic. Specifically, the algorithm conducts intra-segment agglomerative clustering with a preset number of clusters. This preset number is estimated by a pre-defined speaking rate. As this preset number of syllables may be larger than the number of segments, a post-hoc merging process merges adjacent segments with cosine similarity higher than a threshold, which we call the merge threshold. We use a more sensitive segmentation configuration than the original setting by Cho et al. (2024b) to prevent loss of speech contents due to overly broad segments. Specifically, we halve the estimated syllable duration from 200ms to 100ms to cover speech with fast speaking rate, and increase the merge threshold from 0.3 to 0.4. Note that the scores of SDHuBERT in Table 1 are those reported by Cho et al. (2024b), which use the original setting.

**Non-speech Frames** The non-speech frames are initially defined as"knocked out" frames by norm thresholding with SDHuBERT. However, we find that SDHuBERT is still sensitive to non-speech noise events. Therefore, we mark segments where the average absolute amplitude of z-scored waveform is lower than 0.1 as non-speech frames as well.

**Thresholds in Sylber greedy segmentation algorithm** Unlike the thresholds in SDHuBERT, which are heuristically driven, we aim to set the thresholds in our algorithm in a more principled way, especially in the second stage of training where the target segments are dynamically generated. We first set the norm threshold to be optimal boundary between signal (speech) and noise (non-speech), where the likelihoods of being signal and noise are equal. Specifically, we assume both signal and noise distributions to be Gaussian and solve the equality condition. After the first training stage, we

use the pseudo-ground truth segments used for training to get the distribution of segment norms and norms of non-speech frames in the dev split of LibriSpeech. To make the distribution reflect noise, we apply the noise augmentation as described in the denoising objective (Section A.1.3) to each sample. In the second training stage, we update the mean and variance of noise distribution using the non-segment portions of student outputs using an exponential moving average with a decay rate of 0.9999, while keeping the signal distribution the same as initially set. This results in the threshold of 3.09 after training.

On the other hand, we still remain largely heuristically driven in terms of setting our merge threshold. We use a particularly high threshold of 0.8 compared to 0.3 in the previous works. Such a high threshold for merging is effective in Sylber since the features are much cleaner than SDHuBERT. Instead setting this threshold to a fixed number, we sample a value from $[0.8, 0.9]$ during the second stage training, which is somewhat arbitrarily set after visually inspecting multiple samples. For inference, we select 0.8 as the threshold since it is the lowest number in the range we impose during training. While other threshold values (e.g., 0.7 or 0.9) generally work well, the phoneme recognition experiment in Appendix A.2.4 empirically proves that 0.8 is optimal when thresholds of 0.1 increments are tested (Table 11).

### A.1.8 Syllable Segmentation Evaluation Details

The evaluation protocol for syllable detection and discovery follows the previous studies (Cho et al., 2024b; Komatsu & Shinozaki, 2024). We use the syllable labels of dev and test splits of LibriSpeech created by forced aligned phonemes and syllabication of them. Similar to SDHuBERT, Sylber interleaves silence frames between syllables. Thus, we use the front boundaries of the detected segments as the boundary predictions. Since this may incur some shifts in the detected boundaries, we use the dev split for finding the optimal constant shift to correct the boundaries, following the previous studies (Peng et al., 2023; Cho et al., 2024b; Komatsu & Shinozaki, 2024). The boundary prediction is considered a hit if it falls within a 50 ms tolerance window of the ground truth boundary. The scores are measured by precision (Pr), recall (Re), F1, and an additional R-value (R) that is introduced by Räsänen et al. (2009) to balance hit-rate and oversegmentation in measuring the quality.

For syllable discovery, the segment averaged embeddings are first clustered with 16384 centroids using k-means clustering and then the centroids are merged to 4096 using agglomerative (or hierarchical) clustering. We use the same setting as the previous studies (Peng et al., 2023; Cho et al., 2024b; Komatsu & Shinozaki, 2024) to make a fair comparison, whereas we use a separate clustering setting in the main tokenization experiments. The detected segments are mapped to the ground truth syllables by maximizing the temporal overlap between paired segments and syllables. We then measure the purity terms—cluster purity (CP), which indicates how purely each segment cluster is mapped to a syllable, and syllable purity (SP), which measures the reverse—as well as mutual information (MI). We refer to Section 4.1 of Komatsu & Shinozaki (2024) for formal definitions of these terms.

### A.1.9 Rhyming Word Pairs

For the consonant at the onset, we constrain the difference to be phonologically adjacent: voiced or voiceless sounds (e.g., "**d**own" vs "**t**own"), non-nasal or nasal sounds (e.g., "**b**all" vs "**m**all"), or spatially adjacent pairs (e.g., "**l**est" vs "**r**est"). For the consonant at the coda, we confine the words to have /I/ as the nucleus vowel to minimize different coarticulation pattern induced by different ending consonants. We only consider nasality difference at the coda and we regard voiced and unvoiced consonants the same since voiced-ness is relatively subtle at the coda position. Additionally, we include the "n-ng" contrast. Table 8 shows the full list of word pairs.

### A.1.10 Discriminability Index

Given words at the left and right ends in interpolation, $x_L, x_R \in W$, the probability of being the left word given interpolating factor $\alpha$ is defined as $p(x_L|x_\alpha) = \frac{\text{sim}(x_L, x_\alpha) - \text{offset}_L}{\text{sim}(x_L, x_\alpha) - \text{offset}_L + \text{sim}(x_R, x_\alpha) - \text{offset}_R}$. The probability of being the right word is symmetrically defined. We need to subtract an offset due to the high base similarity, as the words are rhyming pairs, such that $\text{offset}_A = \min_{\alpha \in [0,1]} \text{sim}(x_A, x_\alpha)$. Then the empirical risk can be defined as $L_{\text{Disc}}(q|x_L, x_R) := \mathbb{E}_{\alpha \in [0,1]} \mathbb{1}_{\alpha < q} p(x_R|x_\alpha) + \mathbb{1}_{\alpha \geq q} p(x_L|x_\alpha)$, for the decision boundary at $q \in [0, 1]$. The optimal boundary can be drawn by

Table 8: Rhyming word pairs used in the discriminability task.

| Onset | | | | |
|---|---|---|---|---|
| Voicedness | | | | |
| b-p | v-f | d-t | z-s | g-k |
| bay, pay | vill, fill | down, town | zeal, seal | goal, coal |
| bar, par | vine, fine | dall, tall | zip, sip | gap, cap |
| ban, pan | vault, fault | deen, teen | zig, sig | gain, cane |
| bad, pad | vox, fox | dime, time | zoo, sue | gauge, cage |
| Nasality | | Place | | |
| b-m | d-n | l-r | t-s | |
| ball, mall | dose, nose | lock, rock | tank, sank | |
| bean, mean | dull, null | lane, rain | tale, sale | |
| boon, moon | dine, nine | long, wrong | tip, sip | |
| bost, most | deal, kneal | lest, rest | tell, sell | |
| Coda | | | | |
| g/k-ng | n-ng | d/t-n | b/p-m | |
| pig, ping | thin, thing | kid, kin | trip, trim | |
| sick, sing | bin, bing | seed, seen | deep, deem | |
| dig, ding | sin, sing | chit, chin | sip, seem | |
| click, cling | kin, king | grid, grin | rip, rim | |

minimizing the risk, $\alpha^* = \text{argmin}_{q \in [0,1]} L_{\text{Disc}}(q|x_L, x_R)$. Discriminability index (DI) is then defined as the risk at the optimal boundary, averaged over word pairs:

$$\text{DI} := \frac{1}{|W|} \sum_{x_L, x_R \in W} L_{\text{Disc}}(\alpha^*|x_L, x_R) \tag{1}$$

For the models with frame-level features like MFCC or HuBERT, we use dynamic time warping to find an optimal alignment that maximizes similarity. While some sample pairs are already aligned, we find that additional warping yields better scores. For models with syllabic features (SDHuBERT and Sylber), we average across all speech (or norm thresholded) parts of the features as all samples are monosyllabic, yielding a single embedding per sample. We use cosine similarity to measure similarity between embeddings from samples. For frame-wise SSL models, the best layers with the lowest DIs are chosen.

## A.2  ADDITIONAL EXPERIMENTS AND ANALYSES

### A.2.1  EFFECT OF THE SECOND STAGE TRAINING

To check the effectiveness of the second stage training, we compare syllable detection and discovery metrics between the stage 1 and stage 2 models. As shown in Table 9, we observe some gain after the second stage training, especially in precision of the segmentation.

Table 9: Syllable detection and discovery performance comparison between two stages.

| Model | Syllable Detection | | | | Syllable Discovery | | |
|---|---|---|---|---|---|---|---|
| | Pr↑ | Re↑ | F1↑ | R↑ | SP↑ | CP↑ | MI↑ |
| Sylber-Stage-1 | 73.7 | **69.2** | 71.4 | 75.6 | 63.2 | **43.9** | 5.24 |
| Sylber-Stage-2 | **76.6** | 68.3 | **72.2** | **75.9** | **64.0** | **43.9** | **5.28** |

### A.2.2  EFFECT OF DENOISING OBJECTIVE

As demonstrated in left two panels in Figure 4, the syllabic structures are already highly visible without the denoising objective, indicating that the major learning source is self-segmentation distillation rather than the denoising objective. However, adding the denoising objective significantly

improves robustness; otherwise, the model becomes highly sensitive to noisy audio as shown in the right two panels in Figure 4.

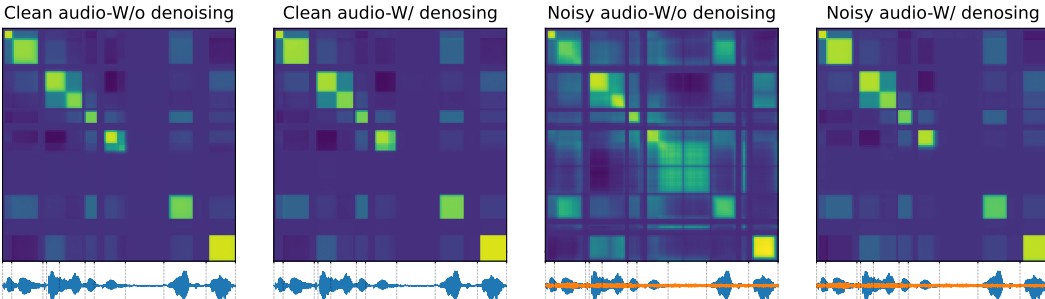

Figure 4: Frame-wise similarity matrix with and without denosing objectives, using clean signal (left two panels) and noisy signal (right two panel). The orange waveform depicts the source noise we add to the clean speech signal.

### A.2.3  CODING EFFICIENCY WITH DURATION-INFORMED TOKENIZATION

When we measure coding efficiency in Section 3, we ignore the duration information. Here, we recalculate the metrics by adding duration as a separate token tagged to each speech token. Note that duration is counted as the number of frames, so it already lies on a discrete space. We find that 99% of HuBERT tokens have duration less than 8, 7, and 6 with the vocab size of 50, 100, and 200, respectively. This means that the duration of each token can be coded by 3 bits. However, when BPE is applied, these 3 bits will be multiplied by the maximum number of units in subwords to count per-token duration bits, which is 10 to 16 depending on the vocab size and cluster granularity.

The syllabic tokens do not densely cover the frames. Therefore, the duration of subsequent silence can be tagged along with the duration of the tokens. 98% of syllabic tokens have duration less than or equal to 16 (4 bits). We can also keep the subsequent silence duration up-to 7 frames (3 bits) efficiently, and silence longer than 7 frames can be regarded as a separate "silence token," adding one more token to the k-means codebook.

Taking all these into consideration, we measure the coding efficiency metrics with the duration-informed tokens as Table 10. Compared to Table 4, the gap between HuBERT-BPE and Sylber gets even larger, where we achieve around or more than $4\times$ gains in bitrate and coding-rate compared to HuBERT baselines. Moreover, even after appending duration tokens, Sylber tokens still have very low bitrates which are below or around 100.

Table 10: Coding efficiency of duration-informed tokens.

| Model | Token/second↓ | | | Bitrate↓ | | | Coding-rate↑ | | |
|---|---|---|---|---|---|---|---|---|---|
| | Vocab size | | | Vocab size | | | Vocab size | | |
| | 5K | 10K | 20K | 5K | 10K | 20K | 5K | 10K | 20K |
| HB50-BPE | 7.45 | 6.82 | 6.30 | 449.26 | 418.24 | 392.36 | 0.0058 | 0.0062 | 0.0066 |
| HB100-BPE | 14.78 | 14.40 | 14.10 | 624.82 | 666.64 | 709.06 | 0.0044 | 0.0041 | 0.0039 |
| HB200-BPE | 16.67 | 15.99 | 15.53 | 654.77 | 979.72 | 967.13 | 0.0043 | 0.0029 | 0.0029 |
| SDHB | 5.84 | | | 112.73 | 118.58 | 124.42 | 0.0239 | 0.0228 | 0.0219 |
| Sylber | 4.76 | | | 91.80 | 96.56 | 101.32 | 0.0297 | 0.0284 | 0.0271 |

### A.2.4  GENERAL REPRESENTATIONAL POWER OF SYLBER

Though the universal utility of our model is not of our focus, we evaluate and benchmark downstream tasks using SUPERB (Yang et al., 2021). First of all, to find the optimal merge threshold, we train a phoneme recognition (PR) model with syllabic embeddings, where the merge threshold is sampled from $[0.3, 0.9]$. The regular CTC based approach is not applicable to syllabic granularity, since it requires that the input length must be no shorter than the target length. Instead, we adopt

RNN-T (Graves, 2012) which does not have a restriction on the length of the sequence. To keep the model size similar to the PR model in SUPERB, we use a very simple, non-RNN transcriber, which is a Layernorm followed by two linear layers where the GELU activation function is applied to the first linear layer's output. The output size of the first layer is set as 768 and set as the vocab size of phonemes, 73, for the second layer. The predictor network has a 3 layer LSTM with a hidden size of 1024, 0.1 dropout rate and Layernorm applied. The model is trained with the RNN-T implementation in PyTorch, and we use beam size of 5 for decoding. The learning rate is set as 0.001 and AdamW is used. The model is trained until no improvement is found in validation loss. We use LibriSpeech clean subsets (train-clean-100, dev-clean, and test-clean), which are the datasets used in the SUPERB PR task setting. For the results in Table 11, the merge threshold of 0.8 is selected and used throughout the SUPERB evaluation. This number coincides with the threshold we use in the main results as well. We use the code provided by S3PRL for the experiment.[8]

Table 11: Phoneme recognition on LibriSpeech (LS) dev-clean with different merge thresholds.

| Dataset | PER $\downarrow$ | | | | |
|---|---|---|---|---|---|
| | Mthr=0.5 | Mthr=0.6 | Mthr=0.7 | Mthr=0.8 | Mthr=0.9 |
| LS dev-clean | 6.15 | 5.88 | 5.73 | **5.68** | 5.68 |

We evaluate 3 versions of Sylber. We freeze the model following the SUPERB protocol.

**Sylber-All Layer** uses all layer features without segmenting with 50 Hz full-sampling rate, being a regular entry to SUPERB.
**Sylber-Segment** uses the segment embedding after segmentation, with syllable granularity.
**Sylber-Segment-Expand** expands segment embedding to the original length.

Table 12 compares these with a HuBERT base model, which has a comparable model size and is trained on the same data. Since Sylber-Segment has a shorter sequence length than the target, thus making the CTC-based recognition task inapplicable, we replace the scores using the aforementioned RNN-T model, and we find a reasonable performance in PR as PER of 5.98, while ASR is lagging behind by a large margin. As our model features are syllabic, this structure may need to be resolved to be converted to characters, adding an additional layer of complexity on top of mapping phonemic features to characters, which is hard to resolve in a limited resource setting.

Another notable point is that our models achieve higher keyword spotting accuracy (KS) and intent classification (IC) compared to the HuBERT base model in all 3 versions. This is aligned with the improved performance in language learning reported in Section 5.3. Also, there is a huge drop in speaker identity detection (SID) when our syllabic embedding is used, indicating that the speaker information is somewhat marginalized out.

Also, the failure in slot filling (SF) and automatic speech verification (ASV) by Sylber-Segment is attributed to the fact that S3PRL is tuned to a lengthy input of speech representation with a regular sampling rate. Further investigation is required, for a proper application of syllabic embedding to those tasks.

Table 12: Performance comparison of various models across different metrics

| Model | PR | KS | IC | SID | ER | ASR | ASR (w/ LM) | QbE | SF | | ASV | SD |
|---|---|---|---|---|---|---|---|---|---|---|---|---|
| | PER$\downarrow$ | Acc$\uparrow$ | Acc$\uparrow$ | Acc$\uparrow$ | Acc$\uparrow$ | WER$\downarrow$ | WER$\downarrow$ | MTWV $\uparrow$ | F1$\uparrow$ | CER$\downarrow$ | EER$\downarrow$ | DER$\downarrow$ |
| Hubert-base | **5.41** | 96.3 | 98.34 | **81.42** | 64.92 | **6.42** | **4.79** | **0.0736** | **88.53** | **25.2** | **5.11** | **5.88** |
| Sylber-All Layer | 11.78 | 96.75 | 98.44 | 76.16 | 64.34 | 11.76 | 8.32 | 0.0623 | 85.79 | 29.21 | 6.72 | 5.08 |
| Sylber-Segment | *5.98 | 97.08 | 98.92 | 50.59 | 64.50 | *14.07 | – | 0.0139 | – | – | – | 13.21 |
| Sylber-Segment-Expand | 88.79 | **97.11** | **99.08** | 51.25 | **65.25** | 12.04 | 8.88 | 0.0591 | 85.66 | 29.49 | 8.75 | 15.55 |

## A.2.5 OUT-OF-DOMAIN GENERALIZABILITY OF SYLBER

To verify whether the syllable segmentation by Sylber can be applied to other domains, we evaluated the model on different datasets from other domains and languages. Specifically, we use the Fisher corpus Cieri et al. (2004), an English conversational dataset with noisy phone call dialogues. As the

---

[8]https://github.com/s3prl/s3prl

training data of Sylber is clean audio-book reading data, evaluation on Fisher can show whether Sylber can work on a different style of speaking and noisy speech. To create the testbed, we sample 200 conversations each for the validation and test sets, and we filtered utterances with less than 3 words spoken, leaving 23K utterances for each set. Furthermore, we also evaluated two datasets from different languages, Spanish and Mandarin, to demonstrate that the syllable boundary detection by Sylber is not limited to English. These two languages are selected since they are the most common languages other than English and a have distinct nature from English. In particular, Mandarin is a tonal language and a distinct Asian language which has a different root from the Indo-European language family. We used a Spanish subset of Multilingual LibriSpeech (MLS) (Pratap et al., 2020) and AISHELL-3 (Shi et al., 2021) for Mandarin.

We follow the same procedure proposed by Peng et al. (2023) and Cho et al. (2024b) to get ground truth syllable segments. We apply the Montreal Forced Aligner (MFA) (McAuliffe et al., 2017) to get phoneme alignments and we group phonemes by syllabification rules to get syllable boundaries. For syllabification, we use a script by Gorman (2013) for English and Silabeador (Sanz-Lázaro) for Spanish. For Mandarin, we regard character boundaries as syllable boundaries since Mandarin is a syllabic language. Figure 9 shows the resulting boundaries. Lastly, we measure the same syllable detection scores as in Table 1 using exactly the same configuration of Sylber and greedy segmentation. Note that we do not include any language or domain-specific optimization or training.

Table 2 shows the boundary detection scores in the three out-of-domain (OOD) datasets. As we can see, all metrics show close to or even better scores compared to the in-domain scores that are denoted in the top row. Sylber shows a surprising generalization capacity in challenging noisy conversation data and even novel languages distinct from English. The similarity matrices in Figure 5 show clean and prominent segments, similar to the in-domain sample shown in Figure 2. This high performance does not require any domain-specific design adaptation. The main reason for this zero-shot multilingual generalization is due to the fact that Sylber represents phonological information rather than other higher order linguistic information like semantic concepts. This finding resonates well with a linguistics perspective that suggests a shared physical basis of phonologies of different languages (Ohala, 1984; 1990). This also corroborates a universal articulatory basis shared across languages, which is revealed by analyzing articulatory physiology encoded in SSL models (Cho et al., 2024a).

Hypothetically, we can use Sylber for initial segmentation of other domain or languages to train a domain-specific or data-scaled domain-general Sylber. To sum, this result suggests a strong potential for our method for real-world, multilingual applications.

### A.2.6    Ablation Experiments: Different Model & Segment Initialization

To demonstrate the robustness of training Sylber, we conduct a set of ablation experiments with different initial segments and model weights. In particular, we simulate a noisier setting by randomly adding noise to the initial segment boundaries. We randomly selected 20% of the syllable boundaries from training, and shifted them between 20-80ms, affecting 36% of the segment annotations in total. This perturbation is applied to the SDHuBERT unsupervised segmentation we used in the first stage and is done once before training.

We train models with two different initialization – SDHuBERT and HuBERT, to see if initialization with SDHuBERT is necessary. We do not include training from scratch since the training is not successful, resulting in degenerate representations. Moreover, we train the models with a reduced setting by reducing the number of updates from 115K to 50K, increasing the learning rate from 1e-4 to 5e-4, and skipping the second-stage training. We measure the same syllable detection and discovery metrics used in Table 1.

As shown in Table 13, the models trained with the noisier initial segments can perform well, showing high scores closer to the main Sylber model. The frame-wise similarity matrices visualized in Figure 6 show similar prominent syllabic structures in both models. The performance difference between two initializations is marginal and no single model outperforms in all metrics. This indicates that HuBERT can also be used for weight initialization and that SDHuBERT is unnecessary if a reasonable initial segmentation is provided.

Furthermore, we train models using phoneme boundaries as initial segments to see the case where the granularity of boundaries is dramatically different. While syllabic segmentation is naturally

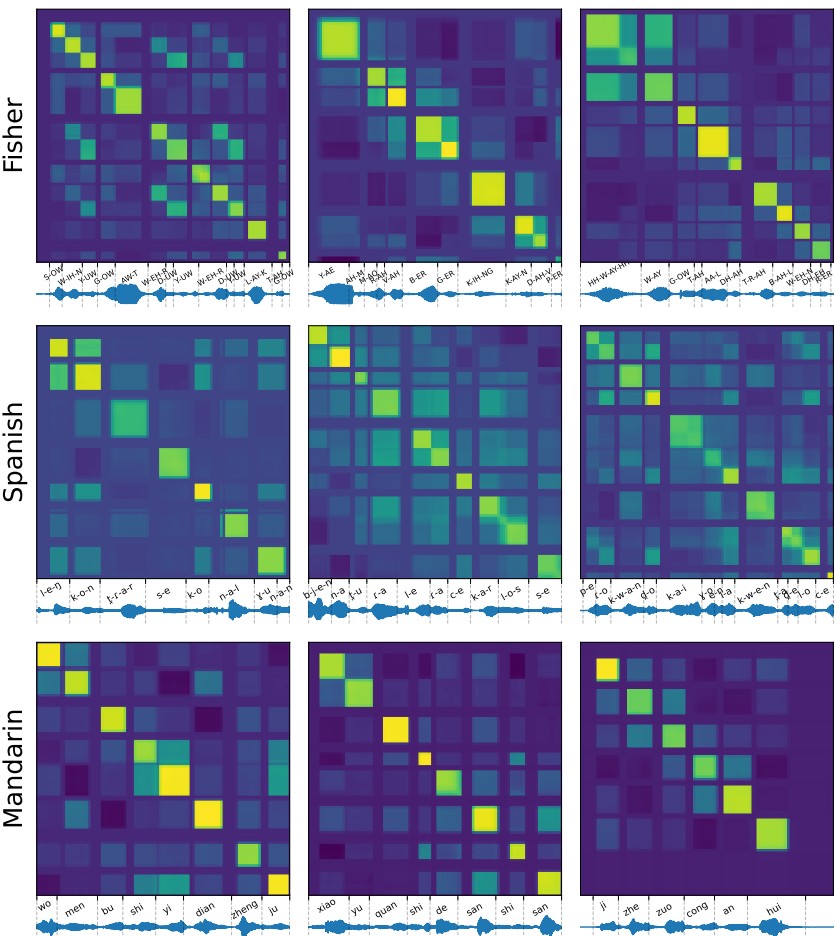

Figure 5: Frame-wise similarity matrices of Sylber applied to samples from OOD datasets: Fisher (top), Spanish (middle), and Mandarin (bottom). The dot product is applied to raw features to measure similarity. We can see highly prominent syllabic segments in all OOD cases.

Table 13: Syllable detection and discovery performance for Sylber trained with noisy initial segmentation with reduced training setting. Original Sylber result is denoted at the top. Pr: precision, Re: recall, R: R-value, SP: syllabic purity, CP: cluster purity, and MI: mutual information.

| Initial Segmentation | Model Init. | Syllable Detection | | | | Syllable Discovery | | |
|---|---|---|---|---|---|---|---|---|
| | | Pr↑ | Re↑ | F1↑ | R↑ | SP↑ | CP↑ | MI↑ |
| SDHuBERT Segment | SDHuBERT | 76.6 | 68.3 | 72.2 | 75.9 | 63.16 | 43.92 | 5.24 |
| Noisy SDHuBERT Segment | SDHuBERT | 74.9 | 67.8 | 71.2 | 75.2 | 61.87 | 42.19 | 5.17 |
| Noisy SDHuBERT Segment | HuBERT | 73.4 | 68.6 | 70.9 | 75.2 | 63.48 | 41.62 | 5.22 |

driven from SDHuBERT, we do not have a readily available unsupervised phonemic segmentation. Therefore, we utilize the ground truth phoneme transcription and alignment inferred by MFA (McAuliffe et al., 2017). We train the exact same settings as above with two different model initializations and the reduced training setting. We measure the same detection metrics but against the ground truth phoneme boundaries (Table 14).

As shown in right two columns in Figure 6, the resulting features are more structured with phonemic granularity, showing prominent squares accurately aligned with the ground truth phoneme boundaries. Moreover, the detection scores are very high, at or over 0.9 (Table 14). Even though the ground truth boundaries are used in training, this is still surprising since sensitivity to boundaries

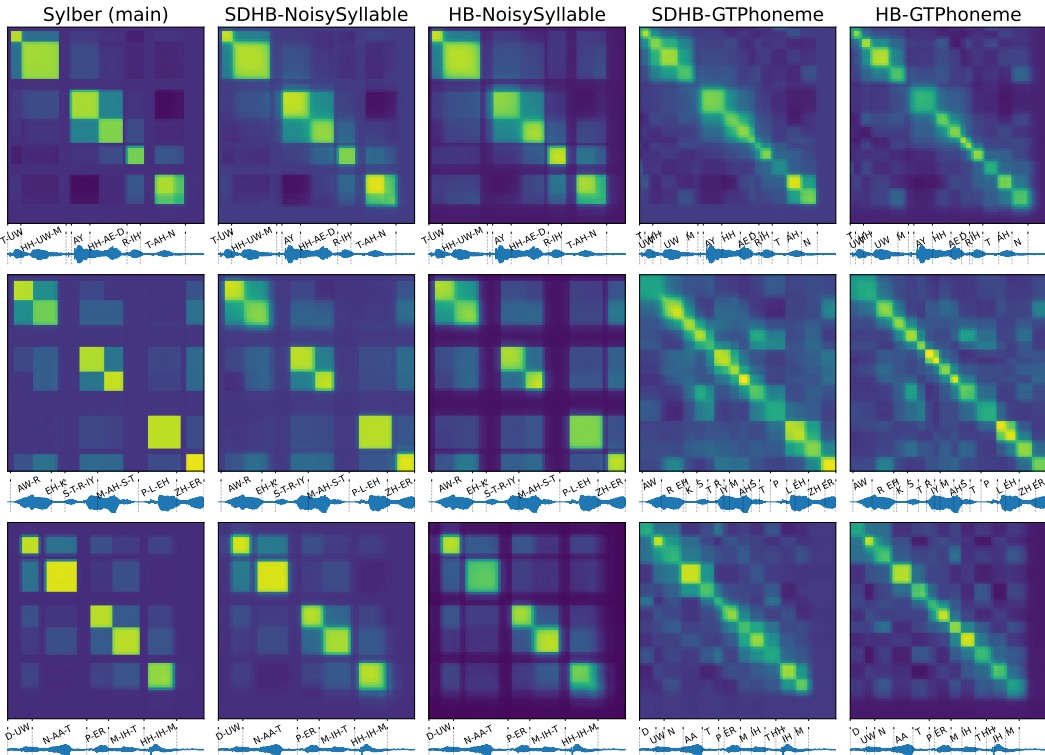

Figure 6: Frame-wise similarity matrix of raw features measured by dot product. Three different samples are shown by rows and columns mean different models. Next to the original model, Sylber (main), the models are denoted as *Initial Model-Initial Segment*. Even with initial noisy syllable segments (NoisySyllable), similar prominent syllable segments emerge from training. When ground truth phoneme boundaries (GTPhoneme) are used, phonemic segments are induced.

Table 14: Phoneme detection performance using ground truth phoneme segments as initial segmentation by different model weight initialization (SDHuBERT or HuBERT). Pr: precision, Re: recall, and R: R-value.

| Initial Segmentation | Model Initialization | Pr↑ | Re↑ | F1↑ | R↑ |
|---|---|---|---|---|---|
| Phoneme Segment | SDHuBERT | 87.6 | 90.3 | 89.0 | 90.4 |
| | HuBERT | **94.2** | **91.6** | **92.9** | **93.6** |

is not guaranteed as the training does not involve any categorization or contrastive objective. Also, unlike the syllable case, the model initialized with HuBERT shows higher performance than the one with SDHuBERT. This indicates that phonemic information is better encoded in HuBERT than SDHuBERT, which aligns with the original motivation behind SDHuBERT.

Lastly, the reduced training setting with higher learning rate is expected to induce a less stable optimization than the original setting. However, we were still able to train models with comparable performance even with additional noise added to the segmentations. This means that our method is not sensitive to a specific choice of hyperparameters, and is easy to train.

### A.2.7 VISUALIZATION OF SYLBER EMBEDDING

To provide a better sense of embedding structure in Sylber, we apply t-SNE to syllabic embeddings obtained from Sylber. We select the 50 most frequent syllables in the LibriSpeech dataset, ignoring stress. Then, for each syllable, we select the top-1K Sylber segments with highest temporal intersection-over-unions with the ground truth syllables and use those embeddings (50K vector em-

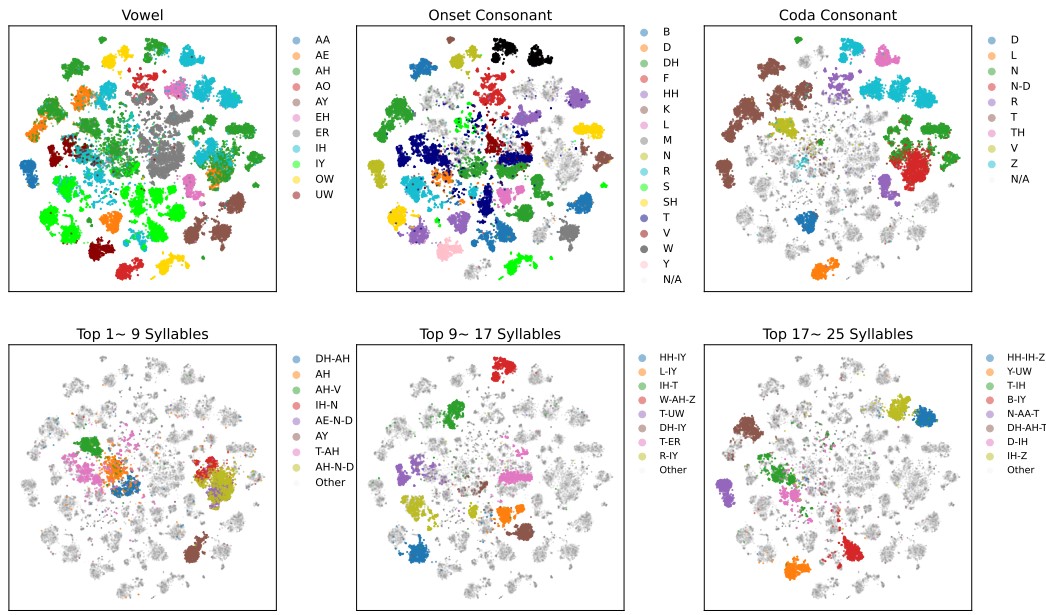

Figure 7: Visualization of Sylber embedding space using t-SNE. The top row shows different colorization by nucleus vowels, onset consonants, and coda consonants. The bottom row shows colorization by different syllables. As shown here, embeddings for each syllable are distinctively clustered, putting similar syllables closer (e.g., /DH-AH/, /T-AH/, and /AH/ in the bottom left panel).

beddings in total). We plot the result with colorization by different categories in Figure 7. Overall, the Sylber embedding space demonstrates a highly discrete structure. The top row shows color distributions by nucleus vowels, onset consonants, and coda consonants. We can see the syllables with the same phoneme components are closely clustered. Also, the distance reflects phonological similarity of the sounds. This is most prominently shown in vowels and coda consonants. For example, the similar vowels /AH/ and /AE/ clusters are adjacent. Furthermore, /N/ and /N-D/ at the coda are clustered together. The bottom row shows color distributions of individual syllables selected from the top-N most frequent syllables. We can see each individual island corresponds to a different syllable suggesting a high level of specificity in Sylber for representing syllables. In addition, the clusters of phonologically similar syllables are adjacent. For example, /DH-AH/, /T-AH/, and /AH/ in the bottom left plot, and /L-IY/ and /DH-IY/ in the bottom middle plot are close to each other. On the other hand, /AE-N-D/ and /AH-N-D/ are not well distinguishable, overlapping in the embedding space, which is natural given the highly phonological similarity between those syllables. In summary, the Sylber embedding space is highly discrete and well aligned with the phonological characteristics of syllables.

### A.2.8 REAL-TIME FACTOR

We evaluate the inference efficiency of Sylber and compare it to SDHubert to show the efficacy of using these models as segmentation models in Table 15. In the first two rows, we evaluate the real-time factor (RTF) in the small batch size regime to measure the efficacy for realtime speech processing. We randomly sampled 32 LibriSpeech files and sequentially ran them through the model and measured the end-to-end latency in order to calculate the RTF. In the latter two rows, we evaluate the RTF in the large batch size regime to measure the efficacy for offline speech processing. We randomly sampled 32 batches of 32 files from LibriSpeech and ran them through the model in a batched manner. All experiments used the same set of randomly selected files in the same order for both SDHuBERT and Sylber. Every experiment was run on a single A6000-48GB GPU with 2 AMD EPYC 7513 32-Core Processor. As a result, Sylber shows a ∼4× reduction in RTF in both the single and batched inference settings compared to SDHuBERT. As we are using a naive numpy implementation for Sylber, this gap can be even larger with a more optimized implementation.

Table 15: Real-time factor (RTF) of syllable segmentation by Sylber and SDHuBERT.

| Batch Size | Model | RTF↓ |
|---|---|---|
| 1 | SDHuBERT | 0.00635 |
| | Sylber | **0.00174** |
| 32 | SDHuBERT | 0.00600 |
| | Sylber | **0.00169** |

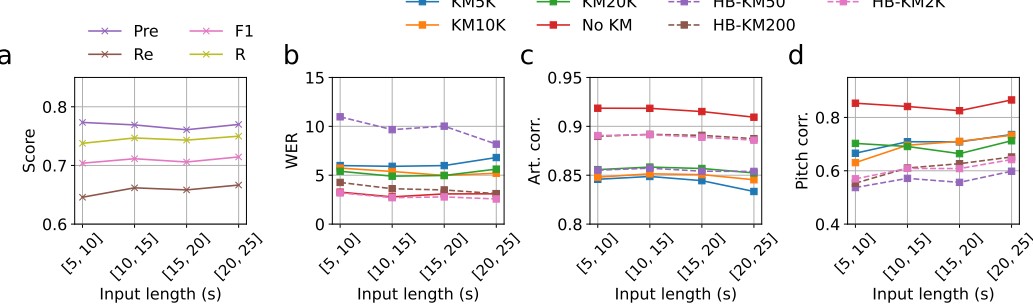

Figure 8: Performance decomposition by different input lengths for Sylber and HuBERT. The x axis shows the input length in seconds. Figure 8a shows Syllable detection metrics while Figures 8b-d show resynthesis scores for different SSL models and unit granularity. The HuBERT units are denoted with 'HB-' in the label and dashed lines. Figure 8b shows WER, Figure 8c shows Articulatory correlation and Figure 8d shows Pitch correlation.

### A.2.9 PERFORMANCE BY INPUT LENGTH

We decomposed the performance for syllable detection and resynthesis into different input length bins. As shown in Figure 8, the syllable detection scores have marginal differences across different input length bins. In terms of resynthesis, there is also minimal difference in input lengths longer than 5 seconds, slightly degrading for 15-20s and 20-25s inputs in resynthesis. This indicates that our token-to-speech model is not dependent on long-context and the speech information is coming from the tokens rather than being inferred from the context. This suggests that our model is not reliant on long-context information; instead, the tokens themselves primarily carry the speech information.

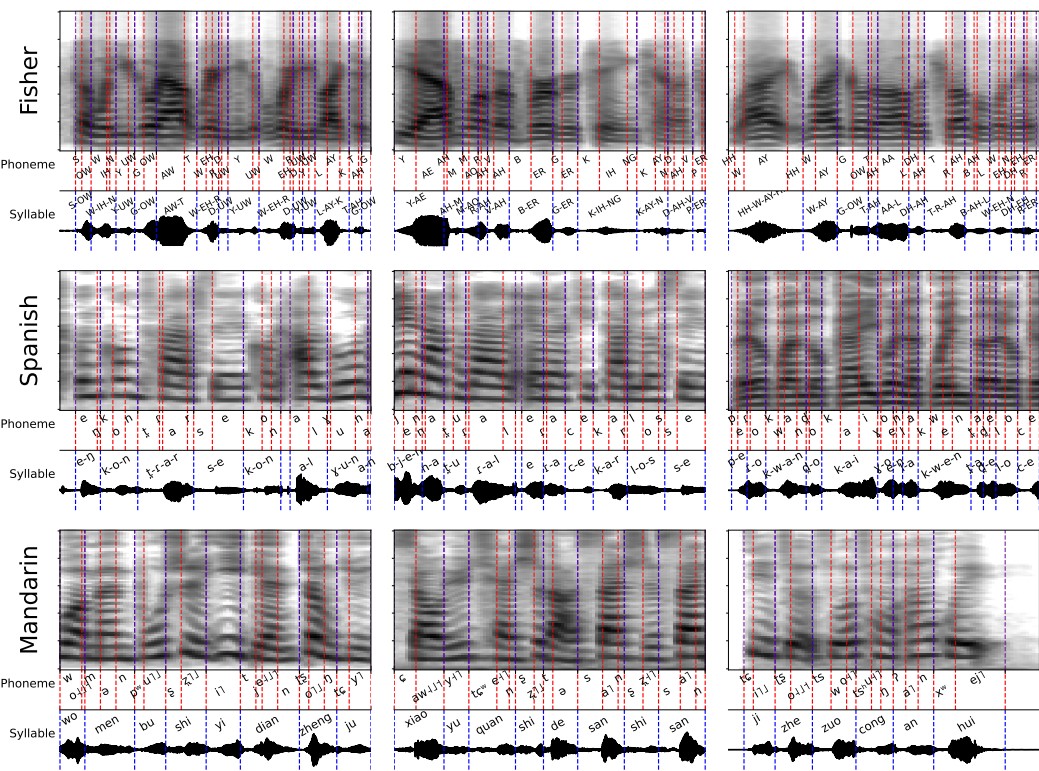

Figure 9: Mel spectrogram with phoneme and syllable alignments on OOD datasets: Fisher (top), Spanish (middle), and Mandarin (bottom). The alignments are obtained by MFA and syllabification to group phonemes into syllables for Fisher and Spanish. For Mandarin, the alignments of characters are regarded as syllable alignments. The Sylber similarity matrices on these samples can be found in Figure 5.

