# OpenReview forum: "Sylber: Syllabic Embedding Representation of Speech from Raw Audio"
_ICLR.cc/2025/Conference — ICLR 2025 Poster_

### Official Review · Reviewer_segj · 2024-10-23

**Soundness:** 3
**Presentation:** 3
**Contribution:** 2
**Rating:** 5
**Confidence:** 4

**Summary:**

This paper proposes Sylber, a self-supervised learning method for extracting syllabic representations from speech. To achieve this goal, the authors propose a self-distillation framework with an unsupervised syllable segmentation algorithm. Sylber surpasses some prior speech tokenizers on multiple benchmarks, including syllable detection, speech resynthesis, and spoken language modeling (understanding). Besides, this paper introduces a discriminability index to measure whether speech embeddings align with categorical perception.

**Strengths:**

1) The proposed Sylber approach successfully bridges the self-distillation method in speech self-supervised learning (SSL) with syllable segmentation/discovery.
2) Sylber demonstrates strong syllable segmentation and discovery capabilities through comprehensive experimental results.
3) The discriminability index and Figure 3 clearly show Sylber's categorical perception capability, which aligns with human perception.

**Weaknesses:**

1) **Training effort:**
It is not clearly stated in the main text, but according to Appendix A.1.4, Sylber requires two-stage training with 1.15M and 500k updates, respectively, even though the model was trained on the 960 hours LibriSpeech. Note that this does not include the pre-training costs of HuBERT and SDHuBERT. Moreover, the performance gain in the second stage of training is not significant (Table 6). E.g., the R-value only improved from 75.6 to 75.9. Thus, the need for the second stage of training is questioned.
2) **Model scalability:**
The unstable training process and sensitivity to hyperparameters are known problems of EMA-based self-distillation, like data2vec. This fact might lead to challenges in scaling the model. Furthermore, larger models increase the training costs, not to mention the required 1.65M updates in the original Sylber model.
3) **Data scalability:**
A commonly known fact between speech datasets is the significant domain differences. According to the experimental results, all models were optimized for LibriSpeech, a relatively clean speech corpus. However, it is unclear whether the segmentation algorithm works in noisier conditions like conversational speech (e.g., Switchboard). Additionally, the segmentation algorithm might need extra design for multilingual speech. Hence, scaling the training data might be difficult, casting doubts about Sylber's real-world applications.
4) **Complicated resynthesis approach:**
Compared to prior studies [2,3], this paper's speech resynthesis (token-to-speech) method is significantly more complicated. It involves a CFM model to generate low-level articulatory features for the Articulatory Encodec. The complicated resynthesis method might introduce more uncertainties to the evaluation results. Also, the resynthesis intelligibility and quality are not significantly better than HuBERT with K-means clustering (Table 2).
5) **Actual inference efficiency:**
The authors only reported the complexity of syllable segmentation methods. However, a more accurate and practical evaluation method is to include the inference time required to extract syllable boundaries. E.g., latency and real-time factors (RTF).
6) **Insignificant SLU improvements:**
The SLU results in Table 4 do not show great improvements over GSLM (less than 3% relative difference). Some prior methods, like NAST [4] and SpeechTokenizer [5], are not reported for comparison. Besides, the dataset for training uLMs is not clearly stated, which is important since uLM performance highly depends on the amount of data used [6].
7) **The necessity of syllabic tokens:**
In addition to the scalability issues previously raised, another question is whether syllabic tokens are necessary in real-world use cases. First, one of the main purposes of developing better speech tokenizers is to advance applications like ASR, TTS, and spoken LM. Nevertheless, the authors only presented a minor improvement in SLU. So, it is unclear whether Sylber helps downstream tasks. Second, because of the loss of fine-grained information (Table 9), tokens extracted from Sylber require incorporating separate speech encoders/tokenizers for more complex problems, which other SSL-based tokenizers or neural audio codecs could address. Perhaps providing convincing reasons and experimental evidence helps justify the need for syllabic tokens.

[1] Chang, Heng-Jui, Alexander H. Liu, and James Glass. "Self-supervised fine-tuning for improved content representations by speaker-invariant clustering." arXiv preprint arXiv:2305.11072 (2023).
[2] Lakhotia, Kushal, et al. "On generative spoken language modeling from raw audio." Transactions of the Association for Computational Linguistics 9 (2021): 1336-1354.
[3] Nguyen, Tu Anh, et al. "Expresso: A benchmark and analysis of discrete expressive speech resynthesis." arXiv preprint arXiv:2308.05725 (2023).
[4] Messica, Shoval, and Yossi Adi. "NAST: Noise Aware Speech Tokenization for Speech Language Models." arXiv preprint arXiv:2406.11037 (2024).
[5] Zhang, Xin, et al. "Speechtokenizer: Unified speech tokenizer for speech large language models." arXiv preprint arXiv:2308.16692 (2023).
[6] Borsos, Zalán, et al. "Audiolm: a language modeling approach to audio generation.(2022)." arXiv preprint arXiv:2209.03143 (2022).

**Questions:**

1) Explain the differences between the discriminability index and phonetic ABX [1,2]. How are these metrics correlated?
2) How many and what kind of GPUs were used to train/fine-tune each model?
3) Why did Table 2 not consider HuBERT with 500 K-means clusters and the continuous representations? Note that 500 clusters or even 2k are commonly used in prior studies [3].
4) Because Sylber was trained with audio in 5-second segments, how does this affect the downstream performance with long utterances (syllable detection and speech resynthesis)?

[1] Schatz, Thomas. ABX-discriminability measures and applications. Diss. Université Paris 6 (UPMC), 2016.
[2] Nguyen, Tu Anh, et al. "The zero resource speech benchmark 2021: Metrics and baselines for unsupervised spoken language modeling." arXiv preprint arXiv:2011.11588 (2020).
[3] Maiti, Soumi, et al. "VoxtLM: Unified Decoder-Only Models for Consolidating Speech Recognition, Synthesis and Speech, Text Continuation Tasks." ICASSP 2024-2024 IEEE International Conference on Acoustics, Speech and Signal Processing (ICASSP). IEEE, 2024.

---

> ### Author Response · Authors · 2024-11-21
>
> We deeply appreciate the reviewer for taking time for reviewing and providing meticulous comments. We conducted additional experiments and analyses to address weaknesses and questions. Below, we address them one by one in detail. We also updated the manuscript accordingly.
>
> **Training efforts (W-1):** The original numbers of updates in the paper, 1.15M and 500K, were mistakenly stated, and the number of updates we actually used are 115K and 50K. We sincerely apologize for the confusion. Therefore the efforts for training are mistakenly overstated. Given this correction, the second stage training is not as big a burden as it originally sounds, and this is even expedited through the proposed linear time greedy algorithm. As the reviewer commented, the second stage does not have a huge effect, but we do not particularly claim that this stage is necessary. Still, the second stage training further refined the segmentation accuracy and quality (especially precision). Most importantly, the second stage training demonstrates an important behavior that self-bootstrapping the segments doesn’t lead to degeneration, and actually can refine the syllabic structure. Therefore, we kept the second stage and used this model for the other experiments and analyses.
>
> **Model scalability (W-2):** We acknowledge the known instability in training with EMA-based self-distillation, especially with a simple regression loss [1]. However, the proposed method is not as sensitive as previous approaches since we initialize the models with SDHuBERT. To further demonstrate this, we additionally trained models with a reduced training setting with two different model initializations, SDHuBERT or HuBERT. In particular, we used a less stable optimization: We decreased EMA decay from 0.9995 to 0.999, increased the learning rate from 1e-4 to 5e-4, and reduced the number of updates from 115K to 50K. We skipped the second stage training given the limited time window. Furthermore, we simulated a noisier setting by randomly adding noise to the initial segments, to make training more challenging. We randomly selected 20% of the boundaries and shifted 20-80ms, affecting 36% of the segment annotations in total. This perturbation is done once before the training.
>
> | Initial Segmentation      | Model Init. | Pr (↑) | Re (↑) | F1 (↑) | R (↑) | SP (↑)  | CP (↑)  | MI (↑)  |
> |---------------------------|-------------|--------|--------|--------|-------|---------|---------|---------|
> | SDHuBERT Segment          | SDHuBERT    | 76.6   | 68.3   | 72.2   | 75.9  | 63.16   | 43.92   | 5.24    |
> | Noisy SDHuBERT Segment    | SDHuBERT    | 74.9   | 67.8   | 71.2   | 75.2  | 61.87   | 42.19   | 5.17    |
> | Noisy SDHuBERT Segment    | HuBERT      | 73.4   | 68.6   | 70.9   | 75.2  | 63.48   | 41.62   | 5.22    |
>
> When syllable detection and discovery scores are measured, the ablation models also show high scores near to the original Sylber (top row), and show similar prominent structures in frame-wise similarity matrices (Figure # in the updated manuscript). This higher learning rate and lower EMA decay is supposed to add instability in training, however, the models can be successfully trained without struggling. This indicates that our method is not as sensitive to hyperparameters as other self-distillation approaches. A proper model initialization is the only prerequisite we have observed since the model is not trainable from scratch. Therefore we expect instability of training would not be a major issue for scaling the model size, though it is yet to be shown whether scaling law is as important in syllable learning as other SSL models.
>
> While scaling the model size does lead to increased costs, this is a problem inherent with the scaling of all SSL based models. Furthermore, this secondary training with Sylber can lead to more efficient and scalable models down the road. We have found that the RTF of running the segmentation using Sylber is so low that it can be done on the fly to segment audio data for further downstream segmentation distillation, so this step of the training process is not a significant bottleneck of the training. (refer to the response below (R5) about RTF or Appendix A.8.) In addition, due to the reduction of the frequency of tokens down to 4.27 tokens/s, this means that spoken language models can train on longer-form speech data more efficiently, since the sequence length required to train these models will be reduced by an order of magnitude, allowing researchers to scalably train SLMs by reducing the amount of memory and GPUs required to scale.

---

> > ### Author Response · Authors · 2024-11-21
> >
> > **Data scalability (W-3):** We appreciate the reviewer for raising this concern. We conducted more experiments, and we found remarkable  zero-shot generalizability to other domains and even unseen languages. We evaluated our unsupervised syllable segmentation on unseen datasets in other domains and languages. Specifically, we used Fisher corpus [2] which consists of noisy phone call dialogues to test the setting suggested by the reviewer. (We couldn’t find annotations for Switchboard, thus we used Fisher instead.) We sampled 200 conversations for the test set, and we filtered utterances with less than 3 words spoken, leaving 23K utterances. We also evaluated two datasets from different languages: Spanish and Mandarin, which are the most common languages other than English. We used a Spanish subset of Multilingual LibriSpeech (MLS) [3] and AISHELL-3 [4] for Mandarin.
> >
> > We followed the same procedure as Cho et al. (2024) [5] to get ground truth syllable segments. We applied Montreal Forced Aligner (MFA) [6] to get phoneme alignments and then we grouped phonemes by syllabification rules to get syllable boundaries. For syllabification, we used a script [7] for English and Silabeador [8] for Spanish. For Mandarin, we regarded character boundaries as syllable boundaries since Mandarin is a syllabic language. Lastly, we measured the same syllable detection scores as Table 1 using the exactly same configuration of Sylber and greedy segmentation. Please note that we do not include any language or domain specific optimization or training.
> >
> > | Corpus                | Language  | Style         | Pr (↑) | Re (↑) | F1 (↑) | R (↑)  |
> > |-----------------------|-----------|---------------|--------|--------|--------|--------|
> > | LibriSpeech (in-domain) | English   | Reading       | 76.6   | 68.3   | 72.2   | 75.9   |
> > | Fisher                | English   | Conversation  | 78.8   | 66.2   | 71.9   | 75.0   |
> > | MLS                   | Spanish   | Reading       | 73.5   | 69.9   | 71.7   | 75.9   |
> > | AISHELL-3             | Mandarin  | Reading       | 74.9   | 68.0   | 71.3   | 75.3   |
> >
> >
> > As a result, we found surprisingly high scores in all metrics, where some scores are comparable to or even surpassing the original in-domain scores (top row). Sylber segmentation works well on noisy audio and OOD style of speaking. This robustness to noise is enabled by the noise augmentation in the training, as  qualitatively shown in Figure 4 in Appendix. Also, the model can successfully segment unseen languages even if they are distinct from English. In particular, Mandarin is a tonal language and has a different root from Latin. The clean syllabic segments that we observed in in-domain samples are also well visible for OOD speech in the frame-wise similarity matrices in Figure 5 in the revised manuscript.
> >
> > These results provide strong evidence that Sylber can generalize to OOD and even other languages. Moreover, this does not require any additional design optimization. In the future, we can use Sylber to segment speech from other domains and languages to train new domain-specific Sylber models. In addition, these results indicate that we may be able to train a multi-lingual Sylber given the out-of-domain generalization of the english-only model to unseen languages. The empirical studies of data and model scaling are left for future directions.

---

> ### Author Response · Authors · 2024-11-21
>
> **Complicated resynthesis approach (W-4):** Our resynthesis model recruited a minimal setting for CFM and we found consistent patterns across different input types as shown in Table 2. In particular, we minimized complications as much as possible by removing input masking and context prompts which were utilized in previous CFM applications in audio or speech [10, 11]. Furthermore, we kept the size of the model moderately small as an 8 layers transformer, and the training configuration to be minimal as we needed to run many different models. Even the randomness is removed at inference to get deterministic output. The target outputs have only 14 dimensions which are significantly smaller than previous CFM applications. In the initial internal exploration phase, we found that a naive regression model that predicts articulatory features directly also works, however, we observed that CFM training is way faster than the regression model. Therefore, we settled down with using CFM for our resynthesis experiments as we need to train multiple models with our limited academic computation resources.
>
> The previous studies brought up by the reviewer use Hifi-GANs, but those models also induce the known instability (e.g., mode collapse) in training GAN and complicated interplays with multiple losses. Therefore, we believe CFM would not specifically involve more complication than the previous methods. In fact, the pattern is highly consistent across HuBERT, SDHuBERT, and Sylber as shown in Table 2.
>
> Regarding using Articulatory Encodec, the encoding and decoding pipeline is fixed and shared across different CFMs, and the previous study shows sufficiently high-performance of this model in terms of synthesizing speech. The decoder is a regular Hifi-GAN which is also utilized in the previous CFM applications where the acoustic features predicted by CFMs are fed to pretrained Hifi-GAN to synthesize audio. The main reason for adopting Articulatory Encodec is that the features are interpretable, so that we can have a better understanding of which and how speech information can be reconstructed from different speech tokens. Such interpretability is not available in other spectral features or VQ-VAE embeddings.
>
> Regarding the performance, the intelligibility using Sylber is comparable to HuBERT units but the advantage is that Sylber units have significantly lower token lengths per second up to 7\~8 times (or 3\~4 times even when BPE applied to HuBERT). As shown in the coding-rate analysis in Table 6, Sylber shows a better trade-off in token lengths and intelligibility than HuBERT. Our listening tests for prosodic appropriateness reveal that, Sylber units show a superior prosody reconstruction with significantly higher pitch accuracy, making Sylber units potentially better at capturing suprasegmental information than HuBERT units. We now include the human subjective evaluation where listeners reported higher prosodic similarity in resynthesis using Sylber (psMOS).
>
> | Model | KM | nMOS (↑) | psMOS (↑) |
>  |---------|---------|---------------|--------------|
> | GT | -- | 4.37 | 4.71 |
> | HB | 200 | 3.24 | 2.65 |
> | HB | 2K | 3.33 | 2.90 |
> | Sylber | 20K | 3.32 | 3.04 |
>  | Sylber | ∞ | **3.80** | **3.62** |
>
>
> We limited the quantization method to K-Means to make a fair comparison and to focus more on demonstrating preliminary results in this new concept of speech representation. However, a simple K-Means would not be an optimal way of quantizing syllable representations as syllables are combinatorial units of phonemes having onset, nucleus, and coda structure (in English). A future study will be conducted seeking a better quantization method that can reflect such structure and achieving the high-performance marked by resynthesis without quantization.

---

> ### Author Response · Authors · 2024-11-21
>
> **Actual inference efficiency (W-5):** We thank the reviewer fo this suggestion. We evaluate the inference efficiency of Sylber and compare it to SDHubert to show the efficacy of using these models as segmentation models in the Table below. In the first two rows, we evaluate the RTF in the small batch size regime to measure the efficacy for real-time speech processing. We randomly sampled 32 LibriSpeech files and sequentially ran them through the model and measured the end-to-end latency in order to calculate the RTF. In the latter two rows, we evaluate the RTF in the large batch size regime to measure the efficacy for offline speech processing. We randomly sampled 32 batches of 32 files of LibriSpeech files and ran them through the model in a batched manner. All experiments used the same set of randomly selected files in the same order for both SDHubert and Sylber. Every experiment was run on a single A6000 GPU with 2 AMD EPYC 7513 32-Core Processor.
>
> | Batch Size | Model     | RTF (↓)       |
> |------------|-----------|---------------|
> | 1          | SDHuBERT  | 0.00635       |
> |            | Sylber    | **0.00174**   |
> | 32         | SDHuBERT  | 0.00600       |
> |            | Sylber    | **0.00169**   |
>
> As we can see, Sylber’s linear time complexity is evident, as it is around 4x faster than SDHubert at segmenting speech.
>
>
> **Insignificant SLU improvements (W-6):** The major gain of using Sylber tokens is in minimizing sequence length and bit-rate. To our best knowledge, speech coding frequency below 5 Tok/s has not yet been demonstrated in SLU. We further experimented with larger scale data to 66K hours of LibriLight. (The original models we reported were only trained on 960 hours of speech) We also added uLM with silence tokens “w/SIL” inserted in the gap between syllables more than 140ms. We were able to achieve a comparable sWUGGY to the model with similar resource settings (small model with cold-init in TWIST [12]). Furthermore, we achieve a sBLIMP score that is slightly better than the largest model with 13B parameters. This is a remarkable improvement given that the token length is below 5 per second and we used only a base small model with 125M parameters.
>
> | Model              | # Param. | Vocab size | Tok/s (↓) | Bitrate (↓) | Corpus | Data size | sWUGGY (↑) | sBLIMP (↑) |
> |--------------------|----------|------------|-----------|-------------|--------|-----------|-------------|-------------|
> | GSLM               | 150M     | 100        | 26.68     | 177.26      | LS     | 1K        | 68.70       | 57.06       |
> | SDHuBERT-uLM       | 125M     | 5K         | 5.24      | 64.39       | LS     | 1K        | 65.80       | 54.87       |
> | SDHuBERT-uLM       | 125M     | 10K        | 5.24      | 69.63       | LS     | 1K        | 67.42       | 54.48       |
> | SDHuBERT-uLM       | 125M     | 20K        | 5.24      | 74.87       | LS     | 1K        | 67.85       | 54.87       |
> | Sylber-uLM           | 125M     | 5K         | **4.27**  | **52.47**   | LS     | 1K        | 67.32       | 57.34       |
> | Sylber-uLM           | 125M     | 10K        | **4.27**  | 56.74       | LS     | 1K        | 68.41       | **58.04**   |
> | Sylber-uLM           | 125M     | 20K        | **4.27**  | 61.01       | LS     | 1K        | **70.27**   | 57.67       |
> |--------------------|----------|------------|-----------|-------------|--------|-----------|-------------|-------------|
> | tGSLM              | 150M     | --         | 5         | --          | LL     | 6K        | 68.53       | 55.31       |
> | NAST               | 150M     | 200        | 28.97     | 221.44      | LL     | 6K        | 76.42       | 55.62       |
> | TWIST-ColdInit     | 125M     | 500        | 16.78     | 150.45      | LL++   | 150K      | 77.74       | 54.27       |
> | TWIST              | 13B      | 500        | 16.78     | 150.45      | LL++   | 150K      | **84.10**   | 59.20       |
> | Sylber-uLM           | 125M     | 20K        | **4.27**  | **61.01**   | LL     | 66K       | 76.31       | 60.54       |
> | Sylber-w/SIL-uLM     | 125M     | 20K        | 4.76      | 68.01       | LL     | 66K       | 78.03       | **60.78**   |
>
> We also included prior models that use tokenizer based on HuBERT or variants, including the suggested NAST. However, we found SpeechTokenizer doesn’t report these metrics so we couldn’t include it.
>
> The uLMs we presented in the original manuscript used LibriSpeech, 960 hours of data, for training. We explained this in Section 5.1: Datasets, which was mentioned altogether with other models which also used LibriSpeech. We now explicitly mention this specifically for uLM to avoid confusions, and the size of training data is added to Table 5. We updated the manuscript accordingly.

---

> ### Author Response · Authors · 2024-11-21
>
> **The necessity of syllabic tokens (W-7):** In addition to the scalability issues previously raised, another question is whether syllabic tokens are necessary in real-world use cases. First, one of the main purposes of developing better speech tokenizers is to advance applications like ASR, TTS, and spoken LM. Nevertheless, the authors only presented a minor improvement in SLU. So, it is unclear whether Sylber helps downstream tasks. Second, because of the loss of fine-grained information (Table 9), tokens extracted from Sylber require incorporating separate speech encoders/tokenizers for more complex problems, which other SSL-based tokenizers or neural audio codecs could address. Perhaps providing convincing reasons and experimental evidence helps justify the need for syllabic tokens.
>
> As we explained in the previous question, we achieved significant improvement in token length and coding efficiency with a consistent comparable performance with previous models that operate on inefficiently lengthy inputs. This reduction in length holds a strong potential in improving efficiency of other downstream models, especially those that utilize large Transformer architecture, where the complexity of the attention mechanism scales quadratically with sequence length. Besides, the scaling speech model has always been bottlenecked by the length of the inputs, compared to the text-based large model, requiring many GPUs [12, 16, 17, 18]. This is even challenging in an academic setting which doesn’t have enough GPU capacities to fit in long context inputs. Our work shows a great potential for resolving this problem.
>
> For the second point, the loss of fine-grained information is prevalent in SSL-based tokenizers. This is well shown in our experiments on HuBERT, which loses way more prosodic information than Sylber as shown in Table 2. Therefore, it is common in SSL-based tokenization to incorporate separate speech encoders [13, 14, 15], for example, speaker embedding or pitch. However, this is actually a benefit, not a caveat. This means the speech information is disentangled and can be controlled/modeled independently. This disentanglement provides better interpretability and controllability of the model behavior, potentially improving safety and transparency of black-box deep models. It is indeed well known that neural audio codecs have fine-grained acoustic information. However, to our best knowledge, none of them is interpretable, often requiring complex model architectures to deal with multiple levels of codebooks, not to mention the significantly high computing cost due to high sampling rate [19, 20, 21]. These are caveats that existing neural audio codecs struggle with.
>
> In fact, the categorical perception analysis (Figure 3) demonstrates that the discarded information is actually removing perceptually inconsequential variance, thereby symbolically abstracting  the embedding. As a result, Sylber can achieve a significantly better coding efficiency than HuBERT.
>
> As we are proposing a conceptually novel framework, we put our efforts more into demonstrating validity, efficiency, and quality of the proposed tokenization, rather than exhaustively investigating downstream applications.
>
> Lastly, we would like to stress that our work also provides an interesting scientific insight that a syllabic, categorical representation may be a natural representation of speech itself. This is because in all processes from HuBERT to SDHuBERT and to Sylber, every step is self-supervised with minimal inductive bias, but syllables naturally emerge and do not degenerate. Moreover, we are the first to demonstrate that a machine categorical perception of speech arises just as in human perception. Beyond industrial interests/necessity/reasons, we believe these findings hold value in advancing our understanding of the nature of speech representation.

---

> ### Author Response · Authors · 2024-11-21
>
> **Difference between the discriminability index and phonetic ABX (Q-1):** The purpose of the two metrics are very different. The discriminability index is targeted to measure the level of categorical perception, while the phonetic ABX is to evaluate phonetic representation, which doesn’t necessarily show whether the representations are categorical or not. This is because the decision is forced by the ABX task which only requires it to be closer than negative reference, not necessarily be tightly similar to positive reference as shown in Sylber (Figure 3). Also, phonetic ABX doesn’t have a densely sampled continuum of sounds which require a careful simulation to keep other aspects (e.g., pitch or duration) the same between pairs.
>
> **GPUs used for training (Q-2):** For training each model of Sylber, CFM, and uLM on LibriSpeech, we used a single A6000 GPU with 48G RAM. For uLM on LibriLight, we used two of them.
>
> **HuBERT tokens with higher granularity (Q-3):** Thank you for bringing this to our attention. We added resynthesis results using 500 and 2K K-means clusters by Tu Anh et al. (2023) [9], which use the 9th layer of HuBERT-Base model (Table 2 in the updated manuscript). However, we could not find a particularly new pattern by using higher granularity other than adding more refinement in intelligibility. The prosody or pitch reconstruction is still largely broken despite using the fine-grained 2K clusters. This result reiterates the finding by Tu Anh et al. (2023) [9], where they reported huge pitch errors in Hifi-GAN trained on those units, and the demo samples with unnatural pitch. We did not consider testing a continuous representation of HuBERT since that doesn’t have tokenization nor any compression in temporal dimension. The main reason for demonstrating the continuous version of Sylber is not for beating HuBERT, more for suggesting the upper bound limit of the syllabic tokenization of speech. Since Sylber shows very high temporal compression with high correspondence to syllables, the continuous tokens can be still useful and we believe knowing the full capacity of the representation is helpful. We leave a further investigation as a future work towards more effective quantization than the naive K-means clustering.
>
> **Detection and resynthesis performance long speech inputs (Q-4):** We found minimal differences when we checked the performance with different input lengths longer than 5 seconds. The plots have been added to the revised manuscript in Figure 8 for your reference. As shown in the figure, the syllable detection scores have marginal differences across different input length bins. In terms of resynthesis, there is also minimal difference in input lengths longer than 5 seconds, slightly degrading in 15-20s and 20-25s inputs in resynthesis.
>
> If you believe we have addressed the concerns, kindly upgrade your recommendation for this paper. Thank you

---

> ### Author Response · Authors · 2024-11-21
> **Reference**
>
> [1] Baevski et al. Data2vec: A general framework for self-supervised learning in speech, vision and language. ICLR 2022
>
> [2] Cieri et al., The fisher corpus: A resource for the next generations of speech-to-text. In LREC, volume 4, pp. 69–71, 2004
>
> [3] Pratap et al. Mls: A large-scale multilingual dataset for speech research. Interspeech 2020
>
> [4] Shi et al. Aishell-3: A multi-speaker mandarin tts corpus and the baselines. Interspeech 2021
>
> [5] Cho et al. Sd-hubert: Sentence-level self-distillation induces syllabic organization in hubert. ICASSP 2024
>
> [6] McAuliffe et al. Montreal forced aligner: Trainable text-speech alignment using kaldi. Interspeech 2017
>
> [7] Gorman https://github.com/kylebgorman/syllabify
>
> [8] Sanz-Lázaro https://github.com/fsanzl/silabeador
>
> [9] Nguyen, Tu Anh, et al. Expresso: A benchmark and analysis of discrete expressive speech resynthesis. arXiv 2023
>
> [10] Le et al. Voicebox: Text-guided multilingual universal speech generation at scale. NIPS 2024
>
> [11] Vyas et al. Audiobox: Unified audio generation with natural language prompts. arXiv 2023
>
> [12] Hassid et al. Textually pretrained speech language models. NIPS 2024
>
> [13] Kharitonov et al. Text-free prosody-aware generative spoken language modeling. ACL 2022
>
> [14] Polyak et al. Speech resynthesis from discrete disentangled self-supervised representations. Interspeech 2021
>
> [15] Nguyen et al. Spirit-lm: Interleaved spoken and written language model. arXiv preprint arXiv 2024
>
> [16] Baevski et al. wav2vec 2.0: A framework for self-supervised learning of speech representations. NeurIPS 2020
>
> [17] Hsu et al. Hubert: Self-supervised speech representation learning by masked prediction of hidden units. TASLP 2021
>
> [18] Radford et al.  Robust speech recognition via large-scale weak supervision. ICML 2023
>
> [19] Wang et al. Neural codec language models are zero-shot text to speech synthesizers. arXiv  2023
>
> [20] Chen et al. VALL-E 2: Neural Codec Language Models are Human Parity Zero-Shot Text to Speech Synthesizers. arXiv 2024
>
> [21] Song et al. Ella-v: Stable neural codec language modeling with alignment-guided sequence reordering. arXiv 2024

---

> > ### Comment · Reviewer_segj · 2024-11-23
> >
> > I truly appreciate the authors' comprehensive responses.
> >
> > **W-1**: Please modify the paper accordingly.
> >
> > **W-2**: I recommend adding these results to the paper.
> >
> > **W-3**: The results on different corpora seem convincing about the generalizability of Sylber. However, I'm curious about why the R-values across different datasets and languages are similar. Could you double-check if the syllable boundaries obtained by MFA are reasonable? For example, visualize the Mel spectrogram and phoneme/syllable boundaries.
> >
> > **W-6**: The additional results could be added to the paper (appendix).
> >
> > **W-7**: The explanation is reasonable. I would like to see some part of it in the main paper.
> >
> > ---
> >
> > Overall, I am willing to increase the score from 3 to 5, but the authors should provide additional experimental results mentioned above.

---

> > > ### Author Response · Authors · 2024-11-24
> > >
> > > We sincerely appreciate your comments on our responses. For the new results and the error correction, we updated the contents and the revised manuscript is available in the openreview (W1: we fixed A.1.5 Training Details; W-2: A.6 Ablation Experiments with Table 13 and Figure 6; and W-6: Table 5 and 6.3 in the main paper). For W-7, we agree this discussion is important and we will optimize space better and add more on this by moving some of the experimental setup into the appendix.
> > >
> > > For W-3, we visually inspected the Mel spectrogram and the phoneme and syllable alignments by the Montreal Forced Aligner (MFA) as suggested. We found the alignments look reasonable. MFA is a popular choice for phonetic segmentation in these languages, and our spot checking also showed that the alignments look reasonable, which we have included as Figure 9 in the appendix.
> > >
> > > We believe syllable detection to be a robust emergent property of Sylber, as the proposed learning objective induces temporal segregation of acoustic frames into larger meaningful, aggregate units. Each of these temporal chunks belong to distinct context dependent unit, which correlate well with syllables. Given that Sylber performs well at this non-uniform parsing task, it naturally generalizes to conversational speaking style as shown in the results on Fisher datasets. And the additional denoising objective makes the model robust to noise. Regarding cross-lingual generalization, there are indeed some hypothesized linguistic universals across languages that our results seem to corroborate [1, 2]. And we believe such discovery is possible since every training step was self-supervised and free of language specific supervision or bias.
> > >
> > > These are still preliminary as we were only able to explore a limited number of languages due to the constrained time window. We will further explore to scale our model to other languages in our future study. We deeply appreciate the reviewer’s comments and willingness to raise the score.
> > >
> > > [1] Ohala. An ethological perspective on common cross-language utilization of f0 of voice. Phonetica, 41(1):1–16, 1984.
> > > [2] Ohala. There is no interface between phonology and phonetics: a personal view. Journal of phonetics, 18(2):153–171, 1990.

---

### Official Review · Reviewer_KE7u · 2024-11-04

**Soundness:** 3
**Presentation:** 3
**Contribution:** 2
**Rating:** 8
**Confidence:** 4

**Summary:**

This paper proposes Sylber, a self-supervised knowledge distillation and bootstrapping method to learn syllable-level speech features. With better features, a faster segmentation algorithm with O(n) complexity can be utilized (compared to O(n^2) algorithms). Better syllable features also improve downstream performance on speech understanding and codec efficiency.

**Strengths:**

1. The authors presented extensive experiments to demonstrate the strengths of Sylber, covering syllable segmentation, spoken language understanding, and audio codec.
2. The propose learning approach is intuitive and having linear time segmentation algorithm would also greatly facilitate utilization of syllable level features into downstream tasks such as spoken language modeling with syllable level tokens.
3. Authors also presented interesting qualitative analysis in Section 4 connecting syllable representations to the categorical perception in rhymed syllable pairs.

**Weaknesses:**

1. Some details and ablation studies are missing:
    1. The proposed method regresses non-speech frames to zero. What model / method was used to determine whether a frame is speech or non-speech?
    2. The authors claim that better features motivate the design of the linear-time greedy segmentation algorithm. For SD-HuBERT and Sylber, how much does that segmentation algorithm affect the performance? It would be clear if the authors can report Table 1 results with all combinations of (SSL feature, segmentation algorithm).
2. The authors should move the slightly negative results in A.3 to the main paper. It is crucial to discuss not just the advantages but also the caveats of the proposed methods.

**Questions:**

1. In Figure 2, does the heat map show syllable level or frame level similarity matrix for Sylber?
2. How robust is the proposed method with respect to the quality of the initial segmentation (currently segmentation from SDHuBERT) and the initial features (currently SDHuBERT) respectively? If phone segmentation instead of syllable segmentation is provided, does the model learn phone level representation instead?
3. See other questions in the Weakness section above

---

> ### Author Response · Authors · 2024-11-21
>
> We deeply appreciate the reviewer for their thoughtful evaluation of our paper and their constructive comments. We conducted additional experiments and analyses. Below, we address the raised weaknesses and questions in detail. We updated the manuscript accordingly.
>
> **Non-speech frames (W-1.1):** We used non-speech frames obtained from segmentation algorithm proposed by SDHuBERT [1]. SDHuBERT demonstrated that frames near some syllable boundaries are “knocked out” at the later encoder layers, showing near-zero norm values. We leveraged these knocked out frames as non-speech frames by simply thresholding norms of frames following SDHuBERT. Additionally, we found many segments from SDHuBERT are non-speech, showing very low waveform amplitudes. Therefore, we filter out segments which have average absolute waveform amplitudes lower than 0.05 (calculated after z-scoring the entire waveform). This is explained in Appendix A.1.6: Threshold Setting in the revised manuscript.  We added a short version of this explanation in the main Method section 3.1 and appended linkage to the Appendix with more detailed explanations. We thank the reviewer for bringing this to our attention.
>
> **Effect of clean representation of Sylber in segmentation algorithm (W-1.2):** We thank the reviewer for suggesting this ablation experiment, which led to  very interesting and useful findings. We evaluated syllable detection and discovery performance with different combinations of feature models and segmentation algorithms. When we apply the previous algorithm, MinCut, which was used by SDHuBERT, to Sylber, we find very marginal differences (Sylber-Mincut). MinCut is designed to search optimal segments at the cost of computation time. Therefore, this indicates that Sylber features are clean and robust enough to find optimal segments in a greedy manner. In fact, when the greedy algorithm is applied to SDHuBERT which is noisier than Sylber, we find significant drops from the original scores (SDHuBERT-Greedy), emphasizing the much cleaner embeddings in Sylber.
>
> In the case of HuBERT, the differences are less prominent and mixed (HuBERT-Greedy), potentially due to the lack of syllabic structure in HuBERT features. We appended these results to the main Table 1 in the revised manuscript.
>
> | Model | Complexity | Pr (↑) | Re (↑) | F1 (↑) | R (↑) | SP (↑) | CP (↑) | MI (↑) |
> |----------------------|--------------|--------|--------|--------|-------|--------|--------|--------|
> | HuBERT | O(kn²) | 51.4 | 31.4 | 39.0 | 50.1 | 33.1 | 28.4 | 3.54 |
> | SDHuBERT | O(n²/k) | 64.3 | **71.0** | 67.5 | 70.7 | 54.1 | **46.2** | 4.76 |
>  | Sylber | **O(n)** | **76.6** | 68.3 | **72.2** | **75.9** | **64.0** | 43.9 | **5.28** |
>  | Sylber-MinCut | O(n²/k) | 76.8 | 68.1 | 72.2 | 75.8 | 63.9 | 44.0 | 5.29 |
> | HuBERT-Greedy | O(n) | 54.5 | 35.2 | 42.8 | 52.7 | 29.5 | 25.9 | 3.36 |
> | SDHuBERT-Greedy | O(n) | 56.1 | 67.4 | 61.2 | 62.1 | 30.0 | 41.5 | 2.67 |
>
> To share some more technical details, for applying greedy algorithms to other models, we searched for the optimal value of the hyperparameters on the dev-clean of LibriSpeech, maximizing R-value. Unlike Sylber features, we found that the greedy algorithm makes boundary decisions too frequently, severely over-segmenting the frames. To mitigate this, we add a preliminary merge operation before the local adjustment in our greedy algorithm, which merges a segment to the next segment if the segment average features of adjacent segments show high similarity. We use the same merge threshold parameter for determining high similarity. This new operation is also linear in time, and actually, it reduces the computation time by merging spurious boundaries and skipping the local adjustment.

---

> ### Author Response · Authors · 2024-11-21
>
> **Discussion on results in SUPERB (W-2):** We present Sylber as more of a novel “coding” framework of speech grounded in a  linguistic and phonological basis. Unlike general SSL models, many previous speech coding or codec studies rarely report SUPERB or diverse downstream performance. Instead, they put more focus on reconstruction quality and coding-rate given the targeted purpose of efficient coding of speech. While following the convention, we decided to include SUPERB as well since what we are presenting is conceptually very novel; therefore, we tried to include as much information as possible. However, given our major purpose and the limited space, we ended up putting the SUPERB results in the Appendix.
>
> We acknowledge the fact that Sylber shows lower performance in some downstream tasks. However, the main purpose of Sylber is not in improving universal representation of speech, rather is more in interpretable representation with temporal structure, trading off some fine grained information of speech. As we show in the resynthesis results, the speech contents can be reliably restored from Sylber features, indicating that the lost information is more in non- or beyond-phonological abstractions learned in the original HuBERT. This may indicate that such lost abstractions are not compatible with syllable-like phonological abstraction, thus requiring another model on top of Sylber features. We believe such downstream models can be efficiently implemented as the input length can be significantly reduced in Sylber.
>
> We strongly agree that the negative results should be equally presented. We are discussing this in detail in Appendix A.3. We would be happy to move this result to the main pages, but we are afraid that we are not able to do so due to the space limit and the aforementioned low priority of SUPERB experiments. Instead, we extended our discussion of our limitations in the main manuscript, and will take into account the discussion from these reviews into consideration in the final version of the paper.
>
> **Question on heat maps (Q-1):** The heat maps are showing frame-level similarities, without any pooling or manipulation.
>
> **Robustness to initial segmentation (Q-2.a):** We appreciate the reviewer for raising this very insightful question. We now include ablation experiments with different initial segments and models. We simulated a noisier setting by randomly adding noise to the initial segment boundaries. We randomly selected 20% of the boundaries and shifted 20-80ms, affecting 36% of the segment annotations in total. This perturbation is done once before the training. We then trained models with different feature model initialization by SDHuBERT or HuBERT. (We didn’t include training from scratch since the training was not successful.) Due to the limited time window, we trained the models with a reduced setting by decreasing EMA decay from 0.9995 to 0.999, increasing the learning rate from 1e-4 to 5e-4, and reducing the number of updates from 115K to 50K, and skipping the second stage training. We evaluated the syllable detection and discovery metrics.
>
> | Initial Segmentation      | Model Init. | Pr (↑) | Re (↑) | F1 (↑) | R (↑) | SP (↑)  | CP (↑)  | MI (↑)  |
> |---------------------------|-------------|--------|--------|--------|-------|---------|---------|---------|
> | SDHuBERT Segment          | SDHuBERT    | 76.6   | 68.3   | 72.2   | 75.9  | 63.16   | 43.92   | 5.24    |
> | Noisy SDHuBERT Segment    | SDHuBERT    | 74.9   | 67.8   | 71.2   | 75.2  | 61.87   | 42.19   | 5.17    |
> | Noisy SDHuBERT Segment    | HuBERT      | 73.4   | 68.6   | 70.9   | 75.2  | 63.48   | 41.62   | 5.22    |
>
>
> As results, the models trained with the noisier initial segments are able to show high detection and discovery scores closer to the main Sylber model. The performance difference between two initializations is marginal and no single model outperforms in all metrics, indicating that HuBERT can be also used for feature model initialization for self-segmentation distillation. The frame-wise similarity matrices visualized in Figure 6 in the revised manuscript show similar prominent syllabic structures in both models.

---

> > ### Author Response · Authors · 2024-11-21
> >
> > **Using phoneme for initial segments (Q-2.b):** Furthermore, to answer the question of what if phoneme segments are used for initial segmentation, we trained models using phoneme segments. However, while the syllable segmentation is naturally driven from SDHuBERT, we do not have a readily available unsupervised phonemic segmentation. Therefore, we utilized the ground truth phoneme transcription and alignment inferred by Montreal Forced Aligner [2]. We trained the exact same settings as above: two different model initializations and the reduced training setting. We measure the same detection metrics but against the ground truth phoneme boundaries.
> >
> > | Initial Segmentation | Model Initialization | Pr (↑)  | Re (↑)  | F1 (↑)  | R (↑)  |
> > |-----------------------|----------------------|---------|---------|---------|--------|
> > | Phoneme Segment       | SDHuBERT            | 87.6    | 90.3    | 89.0    | 90.4   |
> > |  Phoneme Segment      | HuBERT              | **94.2**| **91.6**| **92.9**| **93.6**|
> >
> > As shown in Figure 6, the resulting features are more structured with phonemic granularity, showing prominent squares in frame-wise similarity matrices falling into the ground truth phoneme boundaries. Moreover, the detection scores are very high, near to or over 0.9. Even though the ground truth boundaries are used in training, this is surprising since sensitivity to boundaries is not guaranteed as we don’t have any contrastive or categorization objective. Also, unlike the syllable case, the model initialized with HuBERT shows higher performance than the one with SDHuBERT, indicating that phonemic information is better encoded in HuBERT than SDHuBERT. This means that with our framework, the initialization somewhat matters if the initial units don't have the information encoded. As shown here, both Hubert and SDHubert can serve as a valid initial feature model for Sylber since both have syllable information, but for phoneme, HuBERT is better initialization because SDHuBERT loses the phoneme information.
> >
> > Lastly, the reduced training setting with higher learning rate and lower EMA decay is actually a less stable optimization setting than the original one. This means that our method is not sensitive to the choice of hyperparameter, and easy to train. All of these results are updated in the revised manuscript in Appendix A.6.
> >
> > If you believe we have addressed the concerns, kindly upgrade your recommendation for this paper. Thank you

---

> ### Comment · Reviewer_KE7u · 2024-11-26
> **Questions addressed. Raised the score**
>
> I thank the authors for the very thorough responses. Theses additional experiments provide many interesting insight. Please include them in the manuscript. I have raised the score to 8.

---

### Official Review · Reviewer_AyQW · 2024-11-04

**Soundness:** 4
**Presentation:** 4
**Contribution:** 4
**Rating:** 8
**Confidence:** 5

**Summary:**

This paper studies the problem of tokenizing speech waveforms into discrete units that are suitable for tasks such as speech language modeling. Current approaches to this problem typically cluster (e.g. with K-means) the intermediate representations of a self-supervised transformer (such as HuBERT or WavLM), then run-length encode these clusterings to derive discrete tokens for training a language model over speech units. Two problems with this approach are that 1) the K-means units have been found to represent phonetic/sub-phonetic information and thus have a very high temporal rate, which makes Transformer-based speech language models difficult to scale from a computational perspective and 2) the units do not capture higher-level linguistic abstractions (e.g. words or morphemes) which could make even higher level abstractions (such as semantics) more difficult to learn, preventing current speech LMs from unlocking emergent abilities such as in-context learning.

The paper builds upon a previous approach from the literature, namely SD-HuBERT. The paper uses SD-HuBERT to extract syllable-like segmentations of speech waveforms (without the need for ground-truth annotations). It then fine-tunes an SD_HuBERT model with a teacher-student knowledge distillation objective, where the teacher is an exponential moving average of the student model whose outputs are average pooled features within each syllable-like segment.

The paper presents experimental results showing state-of-the-art performance on syllable segmentation and clustering, and demonstrates that the learned representations exhibit categorical discrimination ("lest" vs. "rest") that mirrors that of humans. It also shows the units can be used to resynthesize intelligible speech and when used to train a speech unit language model achieve strong performance on standard metrics (sWUGGY/sBLIMP)

**Strengths:**

-The paper investigates an important and timely topic, namely speech tokenization focused on learning linguistically-motivated large granularity units.

-The proposed method for learning the units is simple and effective.

-The experiments are broad, covering categorical perception to syllable segmentation to resynthesis to spoken language understanding tasks with speech LMs

-The experimental results are strong on all tasks evaluated.

**Weaknesses:**

The main weakness from my perspective is that I would have liked to see a more in-depth analysis of the resynthesis results in terms of naturalness. It is expected that when moving from low-level acoustic units to higher level syllable-like units, we may lose a lot of the low-level details that are unnecessary for higher level understanding but are needed to represent highly naturalistic speech. However, when building speech LMs we often want to re-synthesize their outputs so they may be played back to the user (e.g. when using the speech LM as a dialog agent) so it is important to understand how well the proposed units work for that scenario.

**Questions:**

Did the authors conduct any naturalness evaluation of the resynthesized speech from the proposed units? How did it compare to existing approaches (such as using HuBERT K-means units)?

---

> ### Author Response · Authors · 2024-11-21
>
> We deeply appreciate the reviewer for recognizing our contributions and insights on the naturalness of resynthesis. We completely agree and have now conducted a human subjective evaluation to get MOS on naturalness (nMOS). And to be further informative, we also ask listeners to evaluate how naturally the prosody of resynthesized speech resembles the original. We denote this as prosodic similarity MOS or psMOS.
>
> | Model | KM | nMOS (↑) | psMOS (↑) |
>  |---------|---------|---------------|--------------|
> | GT | -- | 4.37 | 4.71 |
> | HB | 200 | 3.24 | 2.65 |
> | HB | 2K | 3.33 | 2.90 |
> | Sylber | 20K | 3.32 | 3.04 |
>  | Sylber | ∞ | **3.80** | **3.62** |
>
> As result, we found superior naturalness in prosodic similarity in resynthesis from quantized units of our model compared to that from HuBERT units. The nMOS is similar in HuBERT with 2000 units and Sylber with 20K units. However, as the HuBERT units require significantly more bitrate, 370 bit/s, while ours only need 61 bit/s. We also find room for improvement when we compare to ours without quantization and the ground truth, which we will explore in future study. If you believe we have addressed the concerns, kindly upgrade your recommendation for this paper. Thank you.

---

> > ### Comment · Reviewer_AyQW · 2024-12-02
> >
> > Thanks for your response. I think this is an important paper and will keep my 8 rating.

---

### Official Review · Reviewer_JT6q · 2024-11-06

**Soundness:** 3
**Presentation:** 3
**Contribution:** 3
**Rating:** 6
**Confidence:** 4

**Summary:**

The paper presents an innovative self-supervised learning (SSL) method that converts speech into syllable-based embeddings. The authors employ a range of evaluation metrics—such as syllable detection and discovery, speech intelligibility, coding efficiency, sWUGGY, and sBLIMP—to demonstrate the effectiveness of their approach. The approach provides a linear-time syllable segmentation algorithm and efficient speech tokenization with an average of 4.27 tokens per second.

**Strengths:**

The framework is well-motivated, particularly due to the efficiency of its tokenization algorithm, which helps manage the exponentially increasing compute costs associated with transformer-based models in downstream tasks.

**Weaknesses:**

While the approach is well-motivated as an efficient alternative for speech tokenization, achieving an average rate of 4.27 tokens per second, the evaluation metrics used don’t fully justify the applicability of these tokens for down stream tasks, as shown in Table 9. It would be beneficial for the authors to moderate some claims, such as:

	•	that syllabic units are better suited for lexical and syntactic understanding
	•	and that these units are better suited for SLU

Instead, the focus could remain on highlighting the promising initial results regarding the efficiency of the speech tokenization and interpretability of the tokens, as further work is needed before demonstrating the superiority of syllable-based tokens in downstream tasks.

I would suggest that the authors add more details in Section 3.1 and be more comprehensive.

Also, I would strongly suggest that the authors simplify some of the very long sentences throughout the paper, e.g.,

"The target segment labels are continuous embeddings averaged across frames within each segment that are found by an unsupervised segmentation algorithm"

**Questions:**

- The model was trained to explicitly detect syllables. Wouldn't it perform the best using the proposed metric for syllable detection and discovery? Would metric be a biased one?

- On the author's observation that the articulatory reconstruction and intelligibility increase with finer clustering granularity,  Does that indicates that speech requires more fine-grained representation to performs well in spoken language understanding?

- The author emphasized that other SSL tokens lack structure. Though, other SSL might not have syllable-based structure, but they have sub-phonemic structure

---

> ### Author Response · Authors · 2024-11-21
> **Responses to weaknesses**
>
> We deeply appreciate the reviewer for their thoughtful evaluation of our paper and their constructive comments. Below, we address the raised weaknesses and questions in detail.
>
> **Moderating claims in SLU improvements (W-1)**: We thank the reviewer for the suggestion. We agree with the reviewer’s point regarding our claims. In the revised manuscript, we toned down the statements by removing “better”, and for the second statement, we made it specific to be lexical and syntactic understanding rather than claiming gains in general SLU – thus leaving “syllabic units are suited for lexical and syntactic understanding”.
>
> We also want to share the new experiments that can consolidate the moderated claim. We conducted more experiments with larger dataset (LibriLight 66K) and we found Sylber outperforms a huge size model in sBLIMP with significantly reduced token lengths, and is comparable in the sWUGGY metric with previous models with comparable model sizes (updated Table 5 in the revised manuscript). We also added uLM with silence tokens “w/SIL” inserted in the gap between syllables more than 140ms.
>
> | Model              | # Param. | Vocab size | Tok/s (↓) | Bitrate (↓) | Corpus | Data size | sWUGGY (↑) | sBLIMP (↑) |
> |--------------------|----------|------------|-----------|-------------|--------|-----------|-------------|-------------|
> | GSLM               | 150M     | 100        | 26.68     | 177.26      | LS     | 1K        | 68.70       | 57.06       |
> | SDHuBERT-uLM       | 125M     | 5K         | 5.24      | 64.39       | LS     | 1K        | 65.80       | 54.87       |
> | SDHuBERT-uLM       | 125M     | 10K        | 5.24      | 69.63       | LS     | 1K        | 67.42       | 54.48       |
> | SDHuBERT-uLM       | 125M     | 20K        | 5.24      | 74.87       | LS     | 1K        | 67.85       | 54.87       |
> | Sylber-uLM           | 125M     | 5K         | **4.27**  | **52.47**   | LS     | 1K        | 67.32       | 57.34       |
> | Sylber-uLM           | 125M     | 10K        | **4.27**  | 56.74       | LS     | 1K        | 68.41       | **58.04**   |
> | Sylber-uLM           | 125M     | 20K        | **4.27**  | 61.01       | LS     | 1K        | **70.27**   | 57.67       |
> |--------------------|----------|------------|-----------|-------------|--------|-----------|-------------|-------------|
> | tGSLM              | 150M     | --         | 5         | --          | LL     | 6K        | 68.53       | 55.31       |
> | NAST               | 150M     | 200        | 28.97     | 221.44      | LL     | 6K        | 76.42       | 55.62       |
> | TWIST-ColdInit     | 125M     | 500        | 16.78     | 150.45      | LL++   | 150K      | 77.74       | 54.27       |
> | TWIST              | 13B      | 500        | 16.78     | 150.45      | LL++   | 150K      | **84.10**   | 59.20       |
> | Sylber-uLM           | 125M     | 20K        | **4.27**  | **61.01**   | LL     | 66K       | 76.31       | 60.54       |
> | Sylber-w/SIL-uLM     | 125M     | 20K        | 4.76      | 68.01       | LL     | 66K       | 78.03       | **60.78**   |
>
> Also, we deeply appreciate the reviewer’s acknowledgement in efficiency and interpretability. Indeed, our major focus here is to present a novel framework, which is more relevant to “coding” studies for speech. With this perspective, we added visualization of the embedding space (Appendix A.7) to reinforce the interpretability of the Sylber embeddings. The plot highlights discreteness and phonological arrangement of proposed embeddings. Other main results we are presenting are largely focused on efficiency and quality of coding. We hope moderating those claims eases the concerns of the reviewer.
>
> **Adding details in Section 3.1 (W-2)**: We thank the reviewer for the suggestion. We added more details to Section 3.1 in the revised manuscript. Due to the space limit, many details of our approach and experiments are still placed in Appendix. We will optimize space better and make this section more comprhenesive by moving some of the experimental setup into the appendix.
>
> **Simplifying long sentences (W-3)**: We thank the reviewer for the suggestion. We will give a thorough check to avoid long sentences.

---

> ### Author Response · Authors · 2024-11-21
> **Responses to questions**
>
> **Potential bias in metrics (Q-1):** We'd like to point out that the entire process is self- or unsupervised from waveforms alone, with no “explicit” bias toward human annotated syllables. We are using unsupervised segmentation from SDHuBERT as initial targets for learning segmentation, and we used self-supervised segments from Sylber itself in the second stage training. As no information about the ground truth is given in any process, all the improvements we reported are self-driven, which applies the same for all the models compared. Therefore, we believe there is a minimal bias in the metrics.
>
> **Requirements of finer-grained representation (Q-2):** As we are shifting from (sub) phonemic tokens to syllabic tokens, the token vocabulary requires a larger vocab size to cover frequently used combinations of phonemic units, to fully preserve intelligible contents. We believe that a sufficient level of intelligibility is a prerequisite for language understanding to emerge. Thus, a fine-grained cluster performs well since it can preserve contents more accurately. (But if length is taken into account, Sylber has much fewer granularity than previous HuBERT tokens as shown in Table 2 & 5). However, this may come with a trade-off as we need more training samples to mitigate a long tail issue with less frequent tokens. We expect a similar scaling law demonstrated in NLP literature can be shown using the proposed syllabic tokens since the temporal granularity is similar to that of subword tokens in text, allowing more efficient and effective scaling.  In the current paper, we did not put much focus on exploring the scaling aspects, and kept the experiments in a minimal setting with a basic dataset. We leave this as an interesting direction for future work.
>
> **Structure in previous SSL models (Q-2):** Indeed, we acknowledge that SSL models have sub-phonemic structures as previous studies have demonstrated. As we explain in the introduction, the ideal tokenization for speech has some desirable properties which can be beneficial: temporal segments and categorical discreteness, which are lacking in current tokenization schemes. As shown by [1, 2], the sub-phonemic structure in SSL would be highly continuous, densely tracing the contextualized or coarticulated phonetic representation. This continuous nature leads to inefficient coding in tokenization as we reported in Table 2 and 3.
>
> [1] Sicherman & Adi. Analysing discrete self supervised speech representation for spoken language modeling. ICASSP 2023
> [2] Cho et al. Evidence of vocal tract articulation in self-supervised learning of speech. ICASSP 2023
>
> If you believe we have addressed the concerns, kindly upgrade your recommendation for this paper. Thank you

---

> ### Author Response · Authors · 2024-11-27
>
> We greatly appreciate for dedicating your time and effort to reviewing our work. In our rebuttal and the revised paper, we have aimed to address all the concerns and suggestions raised in the initial reviews. We believe this has become a much stronger paper through the feedback and improvements suggested. We welcome further discussion on any of these points.
>
> Additionally, we would like to share new interesting results we obtained while resolving other reviewers' comments. The follows are the most surprising results we found worth noting. These are all updated in the revised manuscript.
>
> 1. Zero-shot generalizability of syllabic segmentation to other domain and languages (Appendix A.5)
> 2. Stable training of Sylber even with the presence of noise in initial segment (Appendix A.6)
> 3. Additional human-evaluation by mean opinion score (MOS) revealing higher or similar subjective qualities of Sylber tokens compared to HuBERT based tokens (Table 4.)
>
> If all your questions have been addressed and you find these new results interesting, we kindly ask you to consider updating your scores accordingly.
>
> We sincerely appreciate your thoughtful feedback and support.

---

### Meta-Review · Area_Chair_neWF · 2024-12-21

**Metareview:**

The paper proposes a self-supervised approach to learning speech representations. The approach involves distilling from a running average, and an unsupervised segmentation. The syllable information is more prominent compared to other approaches.

This is one of those rare cases where the author-reviewer discussion is very constructive. With the revision, all reviewers find the approach simple and interesting and the experiments insightful.

**Additional Comments On Reviewer Discussion:**

The discussion mainly focuses on scability and the improvements on SLU tasks. The authors are able to provide additional evidence to strengthen their claims, and the reviewers are happy with the revisions.

---

### Decision · Program_Chairs · 2025-01-22

Accept (Poster)